# Antigen recognition detains CD8+ T cells at the blood-brain barrier and contributes to its breakdown

Sidar Aydin[1,8], Javier Pareja[1,8], Vivianne M. Schallenberg[1], Armelle Klopstein[1], Thomas Gruber [2], Nicolas Page [3], Elisa Bouillet [1], Nicolas Blanchard [4], Roland Liblau[4], Jakob Körbelin [5], Markus Schwaninger [6], Aaron J. Johnson[7], Mirjam Schenk [2], Urban Deutsch[1], Doron Merkler [3] & Britta Engelhardt [1] ✉

Blood-brain barrier (BBB) breakdown and immune cell infiltration into the central nervous system (CNS) are early hallmarks of multiple sclerosis (MS). High numbers of CD8+ T cells are found in MS lesions, and antigen (Ag) presentation at the BBB has been proposed to promote CD8+ T cell entry into the CNS. Here, we show that brain endothelial cells process and cross-present Ag, leading to effector CD8+ T cell differentiation. Under physiological flow in vitro, endothelial Ag presentation prevented CD8+ T cell crawling and diapedesis resulting in brain endothelial cell apoptosis and BBB breakdown. Brain endothelial Ag presentation in vivo was limited due to Ag uptake by CNS-resident macrophages but still reduced motility of Ag-specific CD8+ T cells within CNS microvessels. MHC class I-restricted Ag presentation at the BBB during neuroinflammation thus prohibits CD8+ T cell entry into the CNS and triggers CD8+ T cell-mediated focal BBB breakdown.

Multiple sclerosis (MS) is a neuroinflammatory disease of the central nervous system (CNS) characterized by blood-brain barrier (BBB) breakdown, immune cell infiltration and demyelination[1]. Accumulating evidence points to a key role for CD8+ T cells in MS pathogenesis. The major histocompatibility complex (MHC) class I allele HLA-A*0301 is associated with increased risk of developing MS[2]. In active MS lesions CD8+ T cells outnumber CD4+ T cells and oligoclonal expansion of CD8+ T cells in MS lesions and cerebrospinal fluid (CSF) suggests they recognize CNS antigens (summarized in[3]). Furthermore, CD8+ T cells are associated with axonal damage as their cytolytic granules polarize towards demyelinated axons[4]. Clinical trials also underscore the contribution of lymphocyte subsets other than CD4+ T cells to MS pathogenesis[5,6,7].

Immune cell recruitment to the CNS is controlled by the BBB (summarized in[8]). Our present knowledge of the molecular mechanisms involved in T cell trafficking into the CNS is largely based on CD4+ T cells, while the mechanisms used by CD8+ T cells to enter the CNS in MS are still largely unknown. Emerging evidence shows that CD8+ T cells use cellular and molecular mechanisms distinct from those of CD4+ T cells to cross the BBB[9,8]. Specifically, while the migration of CD4+ T cells across the BBB does not require Ag-specific mechanisms[10], the migration of CNS autoantigen-specific CD8+ T cells across the BBB was suggested to rely on the luminal expression of MHC class I[11], implying that CNS antigens are taken up by brain endothelial cells from the abluminal side, processed and cross-presented to circulating CD8+ T cells on the luminal side.

[1]Theodor Kocher Institute, University of Bern, Bern, Switzerland. [2]Institute of Pathology, Experimental Pathology, University of Bern, Bern, Switzerland. [3]Department of Pathology and Immunology, Division of Clinical Pathology, University and University Hospitals of Geneva, Geneva, Switzerland. [4]Toulouse Institute for infectious and inflammatory diseases, University of Toulouse, CNRS, INSERM, UPS, Toulouse, France. [5]Department of Oncology, Hematology and Bone Marrow Transplantation, University Medical Center Hamburg-Eppendorf, Hamburg, Germany. [6]Institute for Experimental and Clinical Pharmacology and Toxicology, Center of Brain, Behavior and Metabolism, University of Lübeck, Lübeck, Germany. [7]Mayo Clinic Graduate School of Biomedical Sciences, College of Medicine, Mayo Clinic, Rochester, MN, USA. [8]These authors contributed equally: Sidar Aydin, Javier Pareja. ✉e-mail: britta.engelhardt@tki.unibe.ch

This prompted us to examine if BBB endothelial cells can cross-present luminally and abluminally available exogenous antigens on their luminal surface on MHC class I molecules and whether this would facilitate the migration of CD8+ T cells across the BBB in vitro and in vivo. Our observations show that brain endothelial cells can take up, process and present exogenous antigens to CD8+ T cells in vitro and in vivo. In vitro MHC class I-restricted Ag presentation on the BBB led to Ag-specific CD8+ T cell arrest and brain endothelial cell apoptosis. Focal BBB breakdown, as observed in CD8+ T cell-mediated auto-immune neuroinflammation in the ODC-OVA mouse in vivo, was found to be independent of the cytotoxic activity of CD8+ T cells. The efficient uptake of ovalbumin (OVA) by CNS-resident macrophages in the ODC-OVA models resulted in limited Ag uptake and presentation by the BBB that still sufficed to reduce the crawling speed of OVA-specific CD8+ T cells within CNS microvessels. Altogether, our in vitro and in vivo observations indicate that neuroinflammation resulting in MHC class I-restricted presentation of Ag at the BBB may reduce CD8+ T cell entry into the CNS. Moreover, high level Ag presentation at the BBB might trigger CD8+ T cell-mediated focal BBB breakdown.

## Results

### Primary mouse brain microvascular endothelial cells (pMBMECs) present Ag to naïve CD8+ T cells

Making use of primary brain microvascular endothelial cells (pMBMECs) as in vitro model of the BBB we first asked if pMBMECs express the basic molecular components required for Ag presentation. Immunofluorescence (IF) stainings on non-stimulated pMBMECs showed weak cell surface staining for MHC class I and no staining for CD80, CD86 or PD-L1. 24 and 48 h of TNF-α/IFN-γ stimulation increased staining for MHC class I and PD-L1 but still not for CD80 or CD86 pMBMECs (Fig. 1A, B). Low expression of CD80 and CD86 mRNA could however be detected in pMBMECs with TNF-α/IFN-γ stimulation inducing a 4.33-fold and 3.52-fold upregulated mRNA expression for CD80 and CD86, respectively (Fig. 1C). To explore if this molecular makeup allows TNF-α/IFN-γ stimulated pMBMECs to prime naïve CD8+ T cells we co-cultured naïve OT-I T cells for 24 h with TNF-α/IFN-γ stimulated and antigen (Ag)-peptide-pulsed pMBMECs. Chicken ovalbumin peptide (257-264) SIINFEKL served as the model antigen, while vesicular stomatitis virus (VSV) nucleoprotein peptide (RGYVYQGL), which binds to H-2K[b] but is not recognized by the TCR of the OT-I cells[12] was used as a control in addition to pMBMECs not pulsed with any peptide. Peptide-pulsed bone marrow-derived dendritic cells (BMDCs) served as professional Ag-presenting cell (APC) control. As expected, SIINFEKL-but not VSV-peptide pulsed BMDCs induced activation of the naïve OT-I cells (Fig. 1C). OT-I cells co-cultured with SIINFEKL-pulsed pMBMECs but not with VSV- or no peptide-pulsed pMBMECs were readily activated as shown by increased cell surface expression of CD69, CD25 and CD44 accompanied with the shedding of CD62L (Fig. 1D). Notably, SIINFEKL-pulsed B2M−/− pMBMECs, lacking functional MHC class I expression, did not induce any OT-I cell activation (Fig. 1D). Thus, pMBMECs can prime CD8+ T cells in an MHC class I dependent manner.

Therefore, we next asked if TNF-α/IFN-γ stimulated, SIINFEKL pulsed pMBMECs can also induce OT-I proliferation following their activation. 72 h of co-incubation of naïve OT-I cells with SIINFEKL-pulsed but not with no peptide or VSV -peptide pulsed pMBMECs induced T cell proliferation as assessed by incorporation of bromo-deoxyuridine (BrdU) into the OT-I cells (Fig. 1E). To explore if pMBMECs can take up and degrade exogenous protein antigens, load peptides on their MHC class I molecules and transport them to the surface for Ag presentation, we co-incubated TNF-α/IFN-γ stimulated pMBMECs with the full-length ovalbumin protein and naïve OT-I cells and observed also under these conditions OT-I cell proliferation to a level comparable to SIINFEKL-pulsed pMBMECs. Activated pMBMECs can therefore process exogenous protein antigens for presentation on MHC class I molecules inducing CD8+ T cell activation and proliferation.

To explore if brain endothelial cell Ag presentation induced CD8+ T cell effector functions we determined induction of effector molecule expression in OT-I cells after co-incubation with TNF-α/IFN-γ stimulated pMBMECs pulsed with either SIINFEKL or VSV peptide or loaded with OVA. Flow cytometry analysis for the cytokines TNF-α and IFN-γ, cytotoxic granule molecules perforin and granzyme B (GrB), lysosome-associated membrane glycoprotein (LAMP)−1 and the apoptosis-inducing transmembrane protein fas ligand (FasL) showed upregulation of IFN-γ and GrB in OT-I cells co-incubated with SIINFEKL-pulsed and OVA-loaded pMBMECs already at 48 h of co-incubation (Supplementary Fig. 1). After 72 h, upregulation of TNF-α, perforin, GrB, LAMP-1 and FasL could be detected in OT-I cells co-incubated with SIINFEKL-pulsed and OVA-loaded pMBMECs but not in unpulsed and VSV-peptide pulsed pMBMECs (Fig. 1F). Expression of cytotoxic effector molecules induced effector functions in OT-I cells leading to the disruption of the pMBMEC monolayers as determined by IF staining for endothelial junctions at 72 h of co-incubation (Fig. 1G). To understand the dynamics of OT-I cell – pMBMEC interactions in the presence or absence of cognate Ag over time we investigated the interaction of tdTomato expressing naïve OT-I cells with unpulsed or SIINFEKL-pulsed pMBMECs isolated from VE-cadherin-GFP reporter mice by in vitro live cell imaging. Observing the interaction of naïve OT-I cells with the pMBMECs over 72 h showed that the number of tdTomato+ OT-I cells co-incubated with unpulsed pMBMECs decreased over time while the pMBMEC monolayer remained intact (Supplementary Movie 1). In contrast, OT-I cells co-incubated with SIINFEKL-pulsed pMBMECs increased in size and initiated disruption of the pMBMEC monolayer as visualized by disruption of the VE-cadherin-GFP labeled junctions (Supplementary Movie 1). Simultaneously SIINFEKL-pulsed pMBMECs without OT-I cells remained intact for the entire observation time.

Taken together our data show that cytokine stimulated pMBMECs can take up, process and present exogenous Ag to naïve CD8+ T cells, leading to their priming, activation and proliferation and induction of effector functions, destroying of the pMBMEC monolayer.

### Endothelial Ag presentation impairs the crawling of naïve CD8 T cells under physiological flow in vitro

To understand if brain endothelial Ag presentation to naïve CD8+ T cells can also be observed under flow as it occurs in the blood stream, we studied the interaction of naïve OT-I cells with pMBMECs under physiological flow by in vitro live cell imaging. TNF-α/IFN-γ stimulated WT or B2M−/− pMBMECs were pulsed with either SIINFEKL or VSV peptides or left unpulsed. Low and comparable numbers of naïve OT-I cells arrested on TNF-α/IFN-γ stimulated pMBMECs irrespective of the presence or absence of cognate Ag or functional MHC class I, underscoring that initial arrest of naïve OT-I cells on pMBMEC monolayers under physiological flow was independent of Ag recognition (Fig. 2A, Supplementary Movie 2). However, while after transient arrest a significant fraction of OT-I cells readily detached from the pMBMEC monolayers under most conditions, this was not observed on SIINFEKL-pulsed WT pMBMECs suggesting that endothelial Ag presentation prohibited OT-I cell detachment (Fig. 2B, Supplementary Movie 2). To further explore how endothelial Ag presentation affects the post-arrest behavior of naïve OT-I cells on pMBMECs under flow we quantified the dynamic interaction of the individual OT-I cells on the pMBMEC monolayers by performing a visual frame-by-frame offline analysis of the time-lapse videos. This analysis showed that in the absence of endothelial Ag presentation, most naïve OT-I cells crawled over the pMBMEC monolayer with a minority showing a probing behavior (Fig. 2B). Diapedesis of naïve OT-I cells across the pMBMEC monolayer was rarely observed. In contrast, on SIINFEKL -pulsed pMBMECs, most naïve OT-I cells remained stationary and actively probed with cellular protrusions the pMBMEC surface with only few OT-I cells crawling over the pMBMEC monolayer (Fig. 2B). The few OT-I cells that crawled over SIINFEKL-pulsed pMBMEC showed a reduced crawling distance and

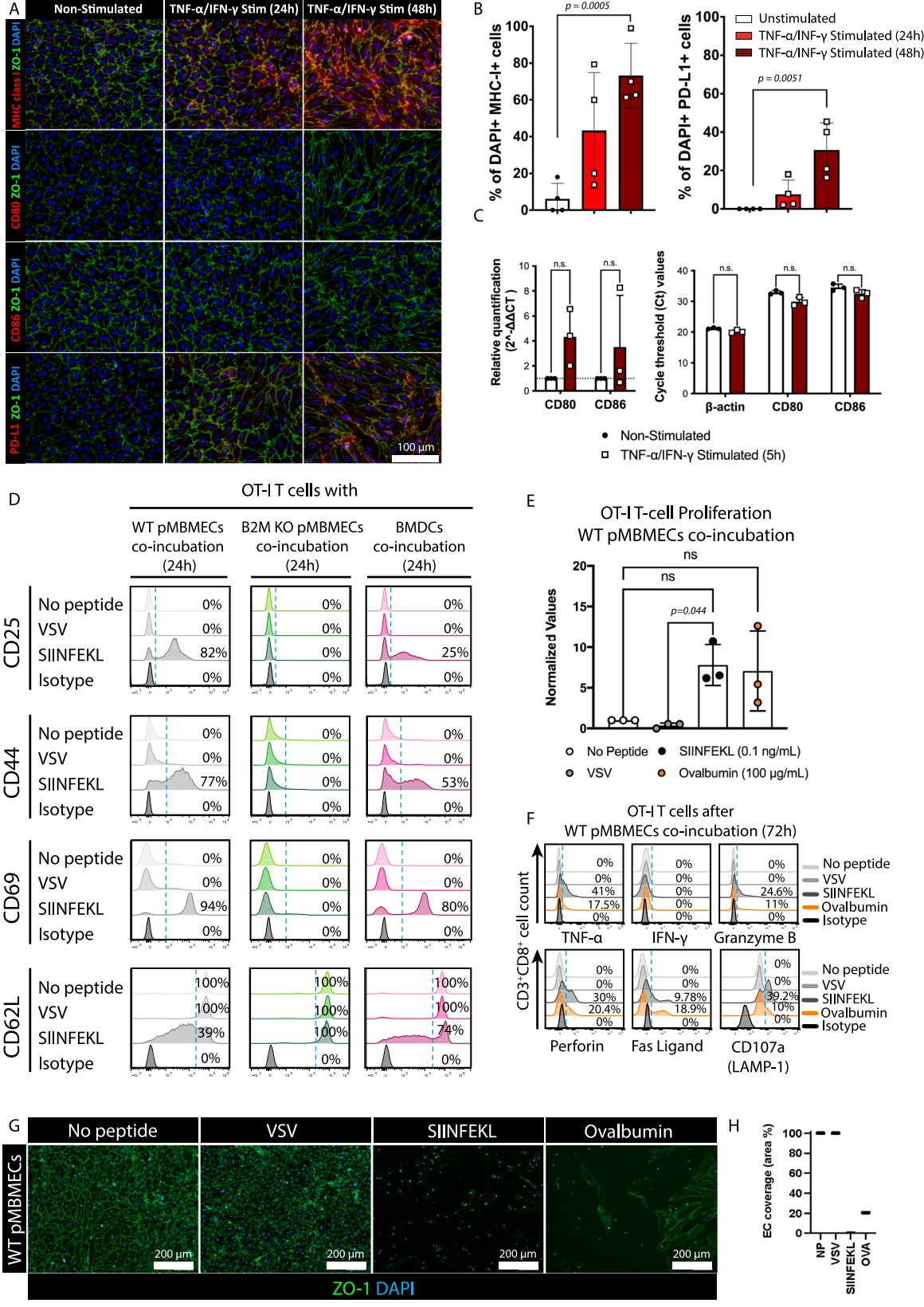

speed when compared to naïve OT-I cells crawling over pMBMECs lacking presentation of the cognate Ag (Fig. 2C, D).

To our surprise, the high avidity of naïve OT-I cell interaction with SIINFEKL-pulsed pMBMECs combined with their probing behavior did not result in their diapedesis across the pMBMEC monolayer (Fig. 2B). Taken together, our observations demonstrate that the arrest of naïve

CD8+ T cells on brain endothelial cells under physiological flow occurs in an Ag-independent manner. Endothelial Ag presentation does however influence their post-arrest behavior seemingly by inducing a stop signal that prohibits naïve CD8+ T cell detachment from and crawling on the pMBMEC monolayer and rather induces their continuous probing of the luminal pMBMEC.

**Fig. 1 | pMBMECs present Ag to naïve CD8+ T cells inducing their activation and proliferation and CD8+ T cell mediated disruption of the barrier in vitro.**
**A** Immunofluorescence staining of pMBMECs for MHC class I, CD80, CD86 and PD-L1, ZO-1, nuclei (DAPI). pMBMECs were stimulated or not with TNF-α/IFN-γ for 24 and 48 h. Staining representative of 5 individual experiments. **B** Quantification of (A) from 4 individual experiments shown as mean ± SD of the percentage of MHC-I+ and PD-L1+ cells over the total number of DAPI+ cells per field of view. Data were analyzed using two-sided unpaired parametric T-Welch's test. **C** Relative gene expression of CD80 and CD86 from unstimulated and TNF-α/IFN-γ -stimulated (5 h) pMBMECs, assessed by qRT-PCR. Technical replicates from 3 individual experiments were measured. Relative quantification represented by the mean ± SD of the 2^−ΔΔCt value. Data were analyzed using two-sided non-parametric Mann–Whitney U test. **D** TNF-α/IFN-γ stimulated pMBMECs from WT or B2M-KO mice or BMDCs were pulsed with SIINFEKL (0.1 ng/mL, 30 min) and co-cultured for 24 h with naïve OT-I T cells. Flow cytometry analysis of OT-I cells after co-culture in the absence or presence of VSV- or SIINFEKL peptides is shown. Histograms depict CD25, CD44, CD69 and CD62L stainings on CD3+CD8+ OT-I cells. Percentage of events above the

dashed blue threshold is indicated. Data represents 3 individual experiments. **E** BrdU-incorporation in OT-I cells after 72 h of co-culture with TNF-α/IFN-γ -stimulated WT pMBMECs in the absence or presence of VSV- or SIINFEKL peptides or full-length ovalbumin (OVA) is shown. Data are pooled from 3 individual experiments with technical replicates, analyzed using one-way ANOVA and shown as mean ± SD normalized to the condition without peptide pulsation. **F** Flow cytometry analysis of OT-I cells after 72 h of co-culture with WT pMBMECs in the absence or presence of VSV- or SIINFEKL peptides or OVA. Histograms show staining of CD3+CD8+ OT-I cells from the co-culture for TNF-α, IFN-γ, granzyme B, perforin, Fas ligand and CD107a (LAMP-1). Percentage of events above the dashed blue threshold is shown. Data represents 3 individual experiments. **G** TNF-α/IFN-γ stimulated WT pMBMECs were pulsed with SIINFEKL or OVA. pMBMECs were co-cultured for 72 h with naïve OT-I cells. Immunofluorescence staining for cell junctions (ZO-1), nuclei (DAPI) is shown. Data represents 3 individual experiments. **H** Quantification of the endothelial cell coverage in (**G**) measured as percentage of the FOV area covered by endothelial cells. Source data from (**B**, **C**, **E**, **F** and **H**) are provided as a Source Data file.

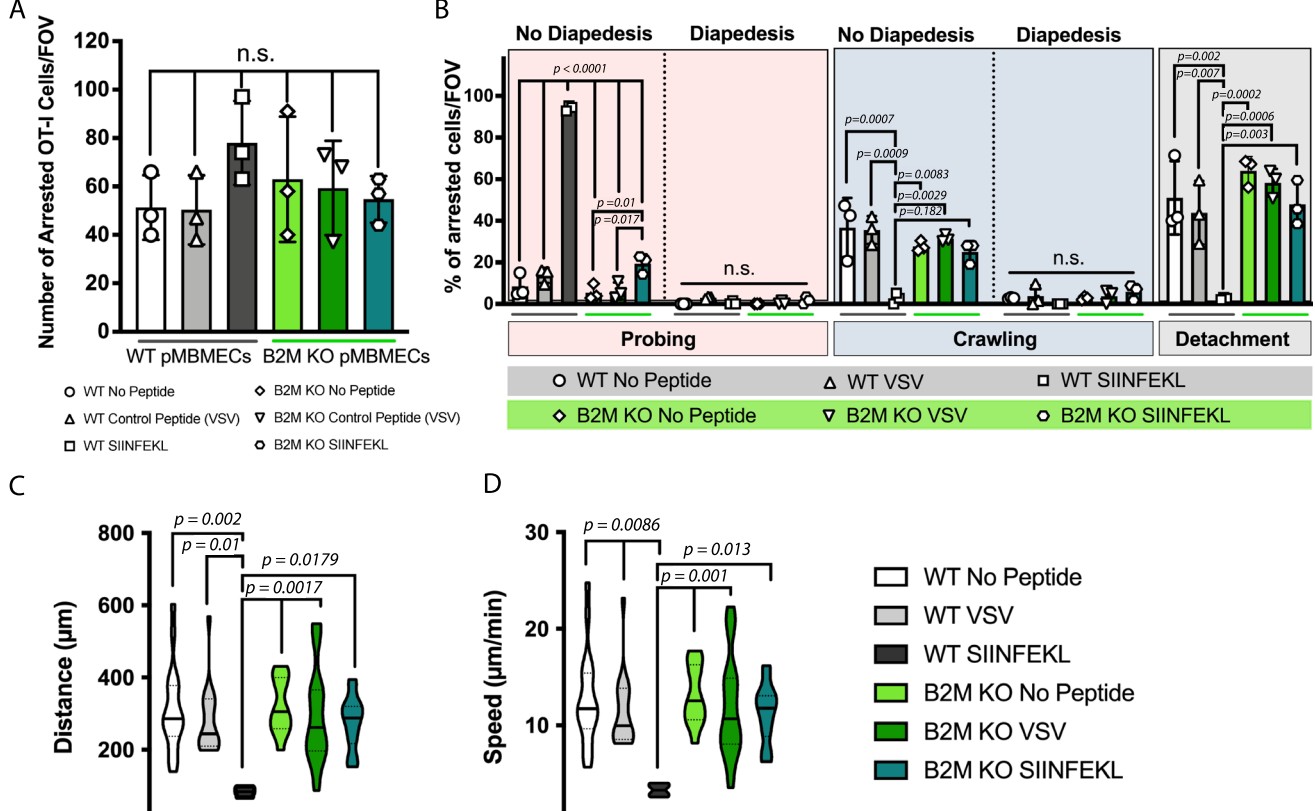

**Fig. 2 | Naïve OT-I cells display impaired crawling on inflamed pMBMECs upon recognition of cognate Ag on MHC class I under physiological flow in vitro.**
**A** Graphs show the number of arrested naïve OT-I cells per FOV (872 × 654 μm) on TNF-α/IFN-γ stimulated WT pMBMECs without peptide (white bar), VSV peptide- (gray bar) and SIINFEKL (black bar) and on B2M-KO pMBMECs without peptide (light green bar), with VSV peptide- (green bar) and SIINFEKL peptide pulsing (teal color bar). Data were pooled from 3 independent experiments, analyzed using ordinary one-way ANOVA with Tukey's multiple comparisons test, and shown as mean ± SD. See also Supplementary Movie 2. **B** Quantification of post-arrest behavior of naïve OT-I cells on WT and B2M-KO pMBMECs during 30 min recording.

The number of arrested naïve OT-I T cells for each condition was set to 100% and the behavioral categories are shown as fraction thereof. Data is shown as mean ± SD from 3 experiments. See also Supplementary Movie 2. Violin plots of the crawling distance in μm (**C**) and speed in μm/min (**D**) of naïve OT-I cells on TNF-α/IFN-γ -stimulated WT and B2M-KO pMBMECs are shown. Values are pooled from three individual expeirments and shown as mean ± SD. For each condition 4–18 cells were tracked. Data were analyzed using ordinary one-way ANOVA with Tukey's multiple comparisons test. Source data from (**A**, **B**, **C** and **D**) are provided as a Source Data file.

## BBB endothelium can process Ags from the abluminal side to prime naïve CD8 T cells at the luminal side

The barrier properties of the BBB prohibit free diffusion of molecules which is modeled by pMBMECs[13]. To explore if BBB endothelial cells could present CNS-derived Ag on their luminal side we next investigated if pMBMECs can take up and process exogenous Ag from their abluminal side and present it on MHC class I on their luminal surface to naïve CD8+ T cells. We grew pMBMECs on filter inserts and after luminal TNF-α/IFN-γ stimulation exposed to increasing concentrations of SIINFEKL or OVA from the abluminal compartment and naïve OT-I cells on the luminal surface. Abluminal exposure of pMBMECs with SIINFEKL or OVA protein induced naïve OT-I cell activation as shown by

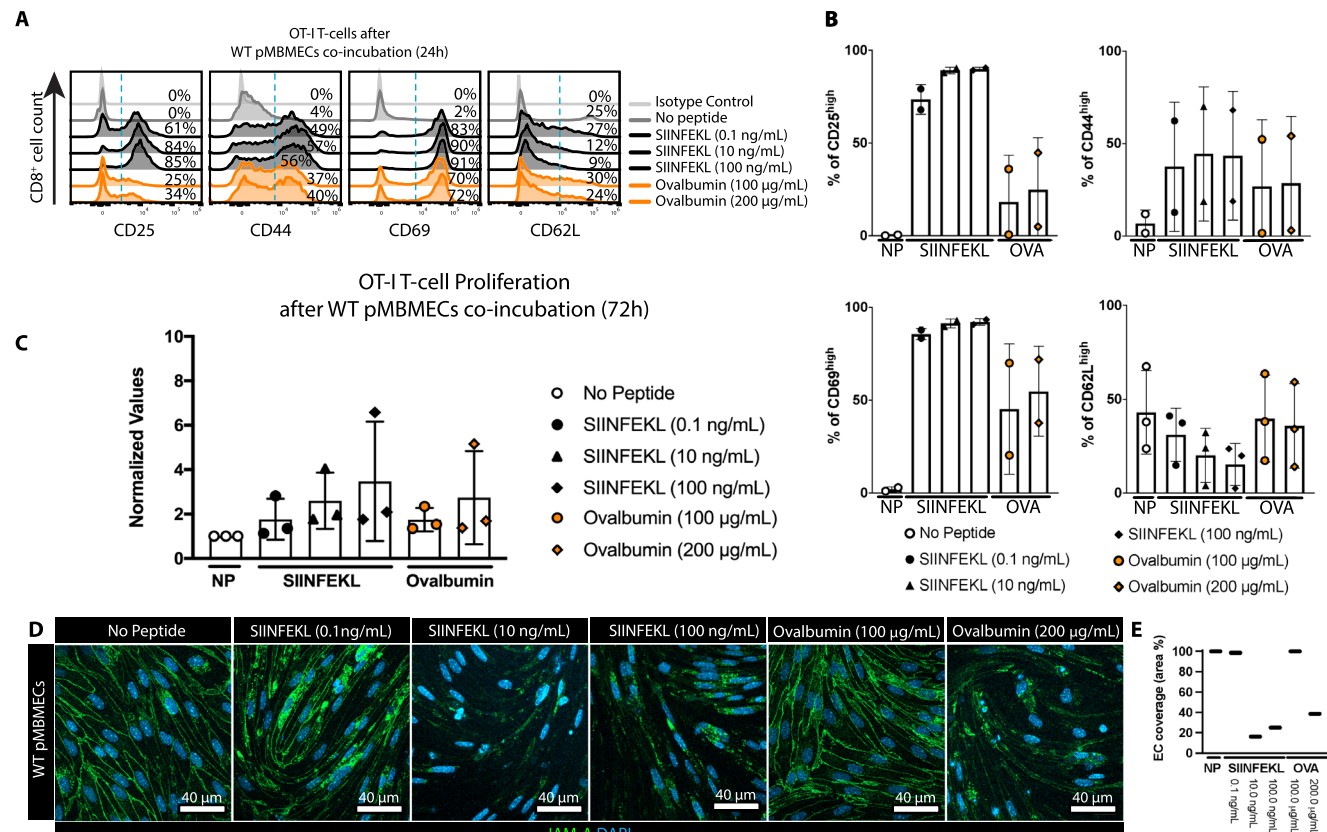

**Fig. 3 | pMBMECs cross-present abluminal Ags to naïve OT-I cells on their luminal side and induce their differentiation into effector cells in vitro. A** Flow cytometry analysis of naïve OT-I cells for activation markers after 24 h of co-culture with TNF-α/IFN-γ-stimulated WT pMBMECs grown on 0.4 µm pore size Transwell filter inserts with abluminal presence of VSV-peptide, SIINFEKL or OVA (SIINFEKL: 0.1 ng/mL, 10 ng/mL, 100 ng/mL; OVA: 100 µg/mL, 200 µg/mL). Histogram plots show staining of CD3+CD8+ naïve OT-I cells after co-incubation for CD25, CD44, CD69 and CD62L. Isotype control for each marker in each condition is shown at the top of each plot. Percentage of events above the dashed blue threshold is indicated. The data is representative of 3 individual experiments. **B** Quantification of (A) represented as percentage of CD8+ cells with high expression of CD25, CD44, CD69 and CD62L. Data were quantified from 2 individual experiments with technical replicates, for CD25, CD44 and CD69 and from 3 individual experiments for CD62L, and shown as mean ± SD. **C** The assessment of naïve OT-I cell-proliferation by BrdU assay after 72 h of co-culture with TNF-α/IFN-γ-stimulated WT pMBMECs on the Transwell filters in the absence of Ag or presence of abluminal SIINFEKL or OVA with increasing concentrations (SIINFEKL: 0.1 ng/mL, 10 ng/mL, 100 ng/mL; OVA: 100 µg/mL, 200 µg/mL). Data were pooled from 3 individual experiments with technical replicates, analyzed using one-way ANOVA and shown as mean ± SD normalized to the condition without peptide. **D** Immunofluorescence staining of cytokine-stimulated WT pMBMECs for JAM-A (green), nuclei (DAPI, blue) after 72 h of co-culture with naïve OT-I cells on Transwell filters in the absence of Ag or presence of abluminal SIINFEKL or OVA (SIINFEKL: 0.1 ng/mL, 10 ng/mL, 100 ng/mL; OVA: 100 µg/mL, 200 µg/mL). Scale bar = 20 µm. The data is representative of 3 individual experiments. **E** Quantification of the endothelial cell coverage in (D) measured as percentage of the FOV area covered by endothelial cells. Source data from (**B**, **C** and **E**) are provided as a Source Data file.

their up-regulation of CD69, CD25 and CD44, and by shedding of CD62L (Supplementary Fig. 2, Fig. 3A). In contrast, OT-I cells cocultured with pMBMECs in the absence of added abluminal Ag remained naïve (Supplementary Fig. 2, Fig. 3A, B). Following their activation, OT-I cells proliferated (Fig. 3C) and differentiated into effector cells leading to the disruption of the pMBMEC monolayers (Fig. 3D).

pMBMECs can thus process exogenous Ag from their abluminal side for presentation on MHC class I molecules on their luminal surface allowing for naïve CD8+ T cell priming and activation and differentiation into effector CD8+ T cells leading to focal BBB breakdown.

### Brain endothelial Ag presentation inhibits effector CD8 T cell migration and initiates CD8 T cell-mediated BBB breakdown

Having shown that MHC class I-restricted Ag recognition on pMBMECs leads to CD8+ T cell effector differentiation and disruption of the pMBMEC monolayer, we next asked how Ag presentation by pMBMECs would affect the interaction with effector CD8+ T cells under shear flow. Investigating the interaction of in vitro activated effector OT-I cells with TNF-α/IFN-γ stimulated WT or B2M−/− pMBMECs under physiological flow by in vitro live cell imaging showed that activated OT-I cells arrested on pMBMECs with approximately sevenfold higher numbers than naïve OT-I cells and that their initial arrest was not affected by the presence or absence of cognate Ag or functional MHC class I (Fig. 4A, Supplementary Movie 3). In the absence of SIINFEKL, effector OT-I cells readily crossed the WT or B2M−/− pMBMEC monolayer following their probing or crawling (Fig. 4B, Supplementary Movie 3). In contrast, on SIINFEKL-pulsed WT pMBMECs OT-I cells showed significantly reduced crawling followed by diapedesis and increased probing behavior without diapedesis (Fig. 4B, Supplementary Movie 3). Also, among the small portion of activated OT-I cells observed to continuously crawl on the pMBMECs, both, the crawling distance and speed were reduced on SIINFEKL-pulsed WT pMBMECs when compared to all other conditions (Fig. 4C, D). It thus seemed that recognition of their cognate Ag on the pMBMECs initiated a stop signal for OT-I cells resulting in their increased probing behavior and their reduced diapedesis upon crawling across the pMBMEC monolayer. To explore if crawling OT-I cells would selectively stop on pMBMECs presenting their cognate Ag, we next investigated OT-I cell interaction

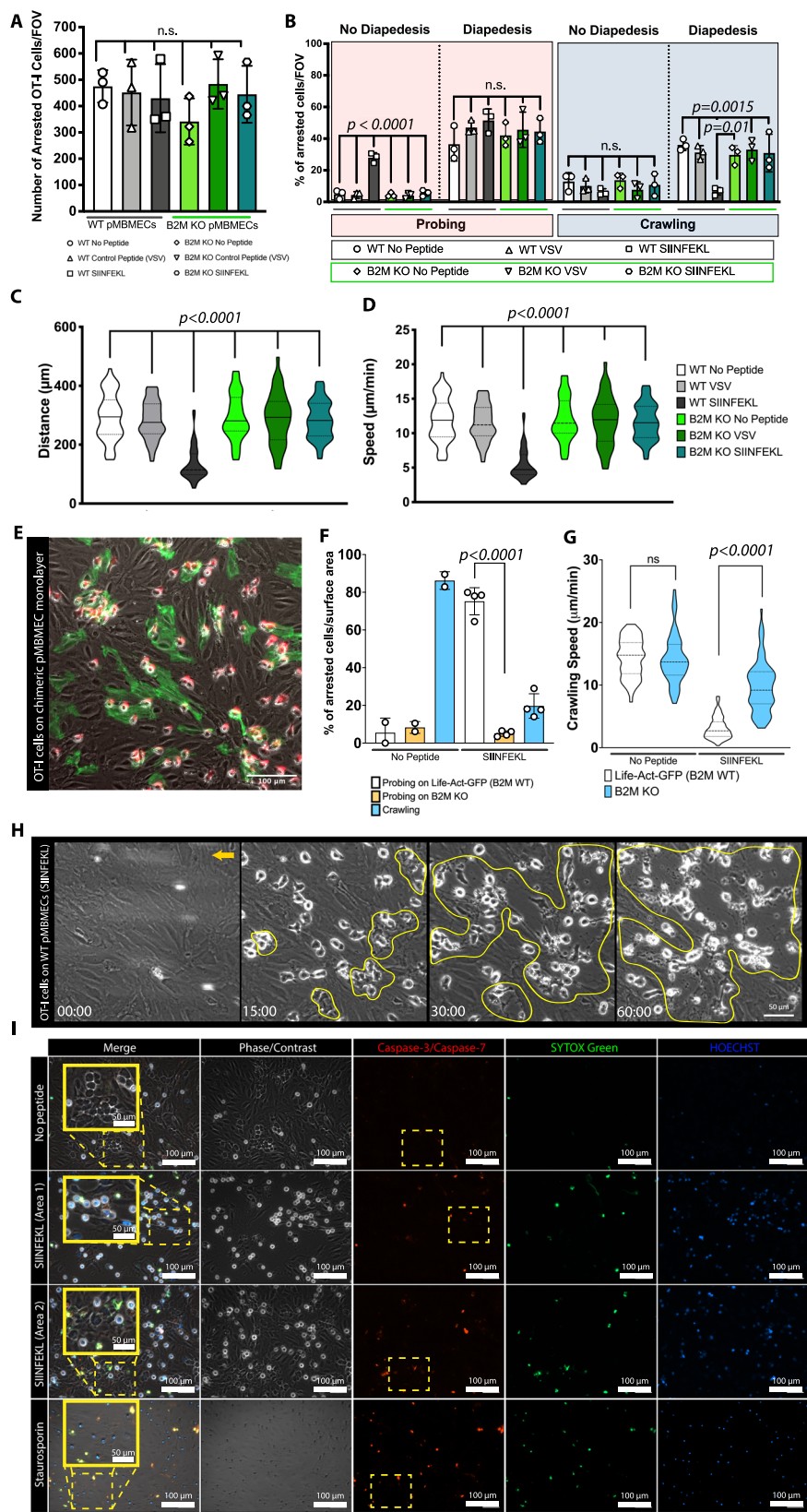

with a heterogenous population of WT and B2M$^{-/-}$ pMBMECs under physiological flow. To this end, we co-cultured pMBMECs isolated from Life-Act-GFP and B2M$^{-/-}$ C57BL/6 J mice that formed a mixed pMBMEC monolayer, where only the GFP$^+$ pMBMECs are competent for MHC class I restricted Ag presentation (Fig. 4E). Investigating the interaction of activated OT-I cells with TNF-α/IFN-γ stimulated mixed

pMBMEC monolayers under flow without prior peptide pulsation revealed no differences in OT-I interaction between MHC class I Ag presentation-competent (Life-Act-GFP$^+$) or -deficient (B2M$^{-/-}$) pMBMECs. By contrast, on TNF-α/IFN-γ stimulated SIINFEKL-pulsed mixed pMBMEC monolayers OT-I T cells were observed to selectively probe over the Ag-presenting GFP$^+$ endothelial cells (Fig. 4F) and

**Fig. 4 | Ag presentation by pMBMECs arrests effector OT-I cells and initiates barrier breakdown under physiological flow in vitro. A** Number of arrested in vitro activated OT-I cells per FOV (872 × 654 µm) on TNF-α/IFN-γ stimulated WT or B2M-KO pMBMECs without peptide, with VSV or SIINFEKL peptide pulsing. Data were pooled from 3 independent experiments, analyzed using ordinary one-way ANOVA with Tukey's multiple comparisons test, and shown as mean ± SD. See Supplementary Movie 3. **B** Post-arrest behavior of effector OT-I cells on WT and B2M-KO pMBMECs over 30 min. The behavioral categories are shown as percentage of arrested OT-I cells on pMBMECs. Data were pooled from 3 independent experiments, analyzed using ordinary one-way ANOVA with Tukey's multiple comparisons test, and shown as mean ± SD. See Supplementary Movie 3. Violin plots for crawling distance (**C**) and crawling speed (**D**) of effector OT-I cells on TNF-α/IFN-γ stimulated WT and B2M-KO pMBMECs. 44-60 cells per condition were tracked. Values were pooled from three individual experiments and are shown as mean ± SD. Data were analyzed using ordinary one-way ANOVA with Tukey's multiple comparisons test. **E** Representative image of effector OT-I T cells superfused over a SIINFEKL-pulsed pMBMEC monolayer composed of Ag presentation competent Life-Act-GFP+ pMBMECs and Ag presentation deficient B2M−/− pMBMECs. Data is representative of 4 individual experiments. See Supplementary Movie 6. **F** Post-arrest behavior of effector OT-I cells on mixed Life-Act-GFP+ and B2M-KO pMBMEC monolayers over 30 min. The behavioral categories are shown as

percentage of arrested OT-I T cells normalized to the surface area of the respective endothelial cell type in the FOV. Data is shown as mean ± SD from 4 independent experiments for the SIINFEKL condition and 2 independent experiments for the No Peptide condition. Data were analyzed using ordinary one-way ANOVA with Tukey's multiple comparisons test. See Supplementary Movie 6. **G** Violin plots for crawling speed of effector OT-I cells on TNF-α/IFN-γ stimulated mixed pMBMEC monolayers. 80-100 OT-I cells per condition were tracked. Data were pooled from 4 independent experiments and analyzed using two-sided paired parametric *t* test. **H** Representative image sequence of effector OT-I cell induced killing of SIINFEKL-pulsed pMBEMCs under physiological flow during 60 min. The area of endothelial cell killing is circled in yellow. Orange arrow indicates the direction of the flow. Data is representative of 3 individual experiments. See Supplementary Movie 4. **I** Live staining of pMBMECs with Image-iT™ LIVE Red Poly Caspases Detection Kit following 60 min of interactions with activated OT-I cells under physiological flow. Unpulsed pMBMECs served as negative control, staurosporin-induced apoptosis as positive control. Gray: phase contrast imaging visualizes the pMBMEC monolayer; Red: Caspase-3/−7 staining for apoptosis, Green: Sytox Green shows cell membrane damage, Blue: HOECHST viusalizes cell nuclei. Areas within dashed yellow boxes are shown magnified in the merged image. Data is representative of 3 individual experiments. Source data from (**A**, **B**, **C**, **D**, **F** and **G**) are provided as a Source Data file.

crawling OT-I cells significantly reduced their crawling speed while crawling over Ag presenting GFP+ pMBMECs, which eventually lead to the destruction of the Life-Act-GFP+ endothelial cells (Fig. 4G and Supplementary Movie 6). In fact, Ag-specific arrest of activated OT-I cells on homogenous WT pMBMEC monolayers under flow resulted in rapid disruption of the pMBMEC monolayer observed already at 15 min after initiation of the interaction (Fig. 4H, Supplementary Movie 4).

To understand if the OT-I cell-mediated destruction of the pMBMEC monolayer is due to the cytotoxic effector activity of OT-I cells we next explored the role of the cytotoxic effector proteins GrB, which was observed to be upregulated in OT-I cells upon in vitro priming (Supplementary Fig. 3). We crossed OT-I mice with GrB deficient C57BL/6 J mice (OT-I GrB−/−) allowing for subsequent side-by-side comparison of the interaction of WT and GrB−/− effector OT-I cells with TNF-α/IFN-γ stimulated, SIINFEKL-pulsed or unpulsed VE-cadherin-GFP pMBMECs under physiological flow by live cell imaging. The newly generated OT-I GrB−/− cells did not show any impairment of their migratory behavior, as we did not observe any difference in the number of arrested cells and their post-arrest behavior between OT-I and OT-I GrB−/− cells (Supplementary Movie 5). However, while upon cognate Ag recognition, effector OT-I cells readily disrupted the pMBMEC monolayer this was not observed during the interactions of effector OT-I GrB−/− cells with pMBMEC under physiological flow (Supplementary Movie 5). Thus, activated OT-I cells can recognize their Ag on MHC class I under physiological flow, leading to the release of cytotoxic granules inducing brain endothelial cell death probably by apoptosis.

To verify if pMBMECs die by OT-I induced apoptosis, unpulsed or SIINFEKL-pulsed pMBMECs were live stained for active caspase activity after one hour of interaction with effector OT-I cells using the fluorescent inhibitor of caspases (FLICA™) methodology. Staurosporine-induced apoptosis of endothelial cells served as positive control. Positive Caspase-3/Caspase-7 staining in combination with membrane damage was observed in SIINFEKL-pulsed but not in unpulsed pMBMECs upon co-culture with effector OT-I cells (Fig. 4I).

Priming of naïve CD8+ T cells into effector or memory CD8+ T cells with a specific Ag is determined by the alterations of numerous cell surface and intracellular molecules[14]. It is, thus, conceivable that in vitro priming of CD8+ T cells may induce effector functions that do not fully resemble those observed in CD8+ T cells activated in vivo[15]. To verify if our in vitro primed CD8+ T cells are a suitable model, we next isolated CD8+ T cells from tdTomato+ OT-I C57BL/6 J mice and activated the OT-I cells in in vitro and in vivo. In vitro priming of naïve tdTomato+ OT-I cells

was performed by a co-culture with professional APCs and the cognate peptide SIINFEKL, whereas for in vivo priming naïve tdTomato+ OT-I cells were injected into WT C57BL/6 J recipient mice 24 h prior to their infection with full length OVA expressing-lymphocytic choriomeningitis virus (LCMV). In vivo and in vitro primed tdTomato+ OT-I cells were found to express comparable cell surface levels for CD62L, CD107a and CD69, whereas in vivo activated OT-I cells showed lower levels of CD25, CD44 and TNF-α (Fig. 5A, B). At the same time, in vivo activated OT-I cells showed a trend towards higher levels of IFN-γ and lower levels of GrB and perforin in comparison to in vitro activated OT-I cells. While in vitro activated OT-I cells showed a homogenous and intense LAMP-1 immunostaining, in vivo activated OT-I cells divided into two subsets with high and low staining intensities (Fig. 5A, B).

As there are subtle differences in the activation profile of in vitro versus in vivo activated OT-I cells, we next aimed to understand if this would impact on the dynamic behavior of OT-I cells on pMBMECs under physiological shear flow. We, therefore, studied the behavior of in vivo activated OT-I cells with TNF-α/IFN-γ stimulated and SIINFEKL-pulsed or unpulsed pMBMECs under physiological flow by in vitro live cell imaging. Lower numbers of in vivo activated OT-I cells arrested on the pMBMEC monolayers when compared to in vitro activated OT-I cells (Figs. 4A and 5C). At the same time in vivo activated OT-I cells did not show any difference in their post-arrest behavior compared to in vitro activated OT-I cells on pMBMECs under flow (Fig. 5C–E). On SIINFEKL-pulsed pMBMECs in vivo primed OT-I cells showed reduced crawling behavior and diapedesis when compared to unpulsed pMBMECs as already observed for in vitro primed OT-I cells (Fig. 5D, E). Moreover, in vivo primed OT-I cells also mediated apoptosis of SIINFEKL-pulsed pMBMECs (Fig. 5F). Overall in vivo activated OT-I effector cells allowed to recapitulate our observations made with the in vitro activated OT-I cells underscoring that upon recognition of their cognate Ag on brain endothelial cells effector CD8+ T cells stop migrating and induce BBB disruption.

pMBMEC cultures always contain rare pericytes even after their negative selection[16]. The Ag presentation capacity of pericytes has not been studied extensively[17]. Therefore, to avoid overlooking Ag presentation by the few pericytes in the pMBMEC cultures, we asked if there is a preferential MHC class I-restricted Ag-dependent interaction of OT-I cells with pericytes rather than pMBMECs. We isolated pMBMECs from NG2-DsRed C57BL/6 J mice to visualize the remaining pericytes in pMBMEC cultures. Observing TNF-α/IFN-γ stimulated, SIINFEKL-pulsed or unpulsed pMBMEC - effector OT-I T cell interactions under physiological flow by live cell imaging we saw that effector OT-I

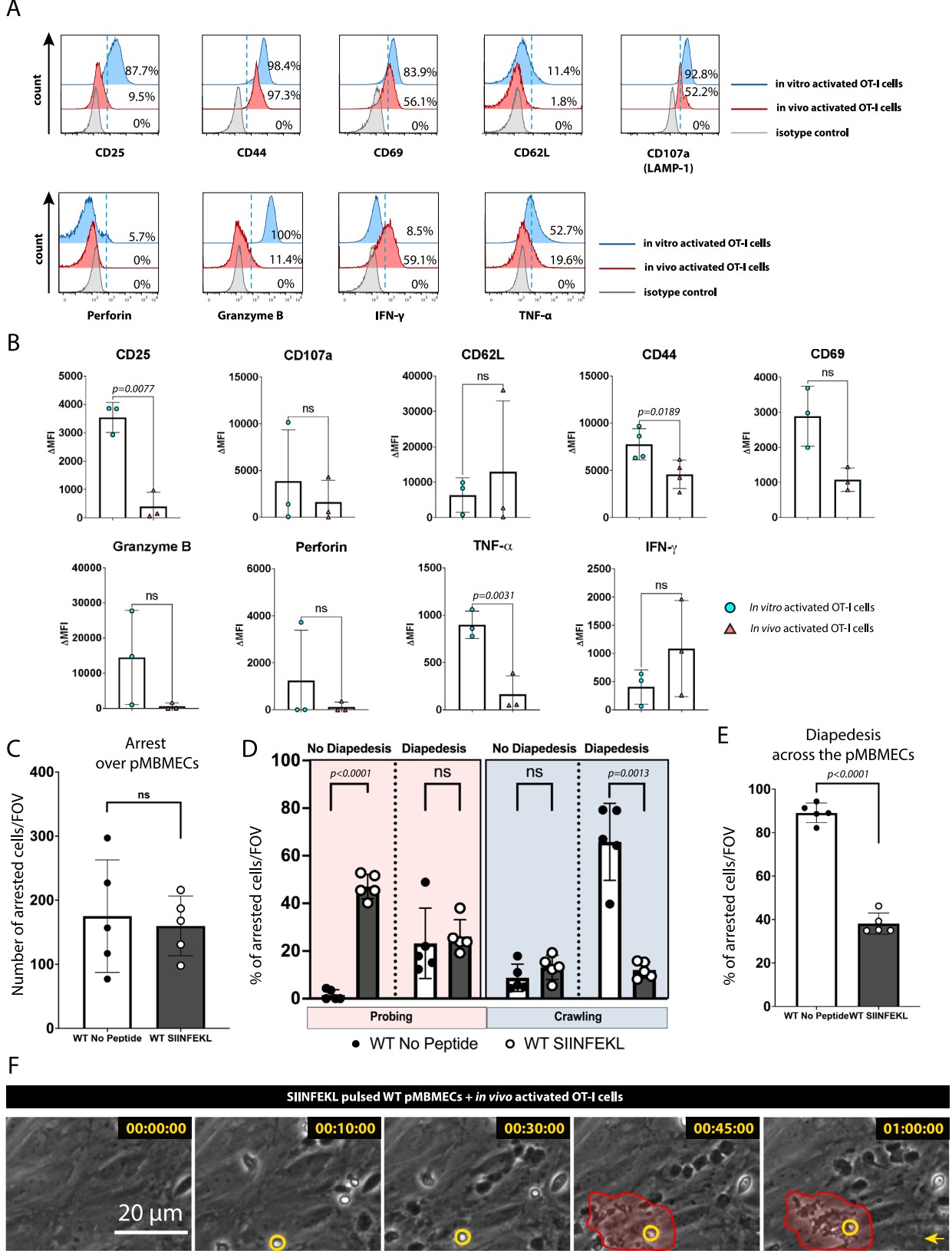

cells continued crawling over the pericytes in the SIINFEKL pulsed-cultures and did neither specifically stop on pericytes nor preferentially induce pericyte death (Supplementary Fig 4).

Taken together, our results show that brain endothelial MHC class I-mediated Ag presentation is not involved in the initial shear resistant CD8[+] T cell arrest on the BBB in vitro. Post-arrest recognition of their cognate Ag on the BBB initiates a stop signal for CD8[+] T cells prohibiting their diapedesis across the BBB and rather promoting CD8[+] T cell mediated disruption of the brain endothelial cells under physiological flow in vitro.

**Fig. 5 | Interaction of in vivo activated OT-I cells with pMBMECs resembles that of in vitro activated OT-I cells.** Naïve tdTomato⁺ OT-I cells were injected into WT C57BL/6 J mice 24 h prior to LCMV-OVA infection. Spleens of the recipient mice were collected at day 8 post-infection and activated OT-I cells were purified by magnetic bead selection and fluorescence-activated cell sorting. **A** Side-by-side comparison of in vitro and in vivo activated OT-I cells. Histograms depict flow cytometry staining for CD25, CD44, CD69, CD62L, TNF-α, IFN-γ, GrB, Perforin and CD107a (LAMP-1) of CD3⁺CD8⁺ single-cell gated in vitro and in vivo activated OT-I cells. Percentage of events above the dashed blue threshold is indicated. Data is representative of 3 individual experiments. **B** Quantification of (**A**) represented as mean ± SD of the mean fluorescence intensity for the respective antigens normalized to their respective isotype controls. Data were pooled from 3 independent experiments and analyzed using two-sided unpaired parametric T-Welch's test. **C**, **D**, **E** Dynamic interaction of in vivo activated OT-I cells with TNF-α/IFN-γ-stimulated pMBMECs under physiological flow in vitro during 30 min of recording is shown. **C** Number of arrested in vivo activated OT-I cells on unpulsed or SIINFEKL-pulsed pMBMECs. **D** Post-arrest behavior of arrested in vivo activated OT-I cells on TNF-α/IFN-γ stimulated pMBMECs under physiological flow in vitro. The behavioral categories are shown as percentage of categorized in vivo activated OT-I cells for each condition on pMBMECs. **E** Bar graph shows the percentage of arrested in vivo activated OT-I cells undergoing diapedesis across the pMBMECs for each condition. Data are representative of three individual experiments and it is shown as mean ± SD. Data were analyzed using two-sided unpaired parametric T-Welch's test. **F** Representative image sequence of in vivo activated- effector OT-I cell induced killing of SIINFEKL-pulsed pMBEMCs under physiological flow during 60 min of imaging. One OT-I CD8⁺ T cell is highlighted by a yellow circle, while the area of pMBMEC killing is marked with red. The yellow arrow indicates the direction of the flow. Data represent 3 individual experiments. Source data from (**B**, **C**, **D** and **E**) are provided as a Source Data file.

## CD8 T cell-mediated autoimmune neuroinflammation increases adhesion of naïve and effector auto-Ag-specific CD8 T cells on the inflamed BBB in vivo

We next asked how brain endothelial cell antigen presentation of CNS antigens may impact CD8⁺ T cell interaction with the BBB in vivo. To this end, we employed the ODC-OVA mouse, which expresses OVA as a neo-self Ag in myelin-forming oligodendrocytes that is solely visible to CD8⁺ T cells but not to CD4⁺ T cells or B cells[18] and thus represents a CD8⁺ T cell mediated mouse model for MS[19]. In this model CNS autoimmune inflammation can be induced by the adoptive transfer of naïve OT-I cells into the recipient ODC-OVA mouse 24 h prior to their infection with LCMV-OVA allowing for in vivo activation of the OT I cells, which will then infiltrate the CNS and cause clinical disease[20]. As a control, adoptive transfer of naïve OT-I cells followed by LCMV-OVA infection was also performed in WT C57BL/6 J mice. As previously reported 5-6 days following LCMV-OVA infection, ODC-OVA mice exhibited neurological signs, accompanied by weight loss, whereas WT control mice did not show any clinical manifestations[21].

To verify that BBB endothelial cells in vivo exhibit the potential to present antigens in an MHC class I restricted manner, we performed double-immunostainings for podocalyxin as a vascular marker and MHC class I and the co-stimulatory molecules CD80, CD86, CD40, VCAM-1 and ICAM-1 on brain and spinal cord sections obtained from ODC-OVA and WT C57BL/6 J mice at day 7 after LCMV-OVA infection (Fig. 6A–G and Supplementary Fig. 5). Additional stainings were performed on brain and spinal cord sections from VE-Cadherin-GFP ODC-OVA and VE-Cadherin-GFP mice expressing a C-terminal GFP fusion protein of VE-cadherin in the endogenous VE-Cadherin locus allowing for direct visualization of the endothelial adherens junctions[22] (Fig. 6F and Supplementary Fig. 5). MHC class I immunostaining strongly colocalized with podocalyxin-positive microvessels in spinal cord and brain gray and white matter in ODC-OVA and WT C57BL/6 J mice indicating a homogeneous expression of MHC class I in the CNS microvessels (Fig. 6A, G and Supplementary Fig. 5). Positive immunostaining for MHC class I on CNS parenchymal cells was additionally observed in ODC-OVA but not WT C57BL/6 J control mice, underscoring the ongoing autoimmune neuroinflammation in the ODC-OVA mice (Fig. 6A and Supplementary Fig. 5). We also observed positive vascular immunostaining for the co-stimulatory molecules CD80, CD86, CD40 as well as for VCAM-1 and ICAM-1 in ODC-OVA and VE-Cadherin-GFP ODC-OVA mice indicating their upregulation under autoimmune neuroinflammatory conditions (Fig. 6B–G). Positive vascular ICAM-1 immunostaining at levels comparable to those observed in ODC-OVA mice was also observed in VE-Cadherin-GFP control mice (Fig. 6F, G), suggesting that even at day 7 post LCMV-OVA infection and complete clearance of the LCMV virus the BBB still retains a mild inflammatory phenotype. In contrast, ICAM-1 immunostaining on CNS parenchymal cells was only observed in the VE-cadherin-GFP ODC-OVA mice but not in the VE-Cadherin-GFP control mice, again underscoring the ongoing autoimmune neuroinflammation in the ODC-OVA background (Fig. 6B).

Having confirmed that inflamed CNS endothelial cells express MHC class I and key co-stimulatory molecules we next asked if this translated into visible Ag-specific changes in the dynamic interactions of CD8⁺ T cells with the BBB in vivo. We therefore induced autoimmune neuroinflammation by the transfer of naïve OT-I T cells 24 h prior to LCMV-OVA infection into ODC-OVA and as well as in WT C57BL/6 J control mice (Fig. 7A). On day 7 after LCMV-OVA infection we observed the interaction of systemically injected CMFDA - labeled naïve or effector OT-I cells in spinal cord microvessels by real time epifluorescence intravital microscopy (eIVM) (Fig. 7A, B). While in WT C57BL/6 J mice we hardly observed any interaction of naïve OT-I cells with the spinal cord microvessels, few naïve OT-I cells transiently adhered in spinal cord microvessels in the ODC-OVA mice (Fig. 7C). When compared to naïve OT-I cells, effector OT-I cells adhered in higher numbers on the inflamed spinal cord microvessels and showed sustained adhesion over time (Fig. 7C, D). Observing visibly higher numbers of OT-I cells adhering in spinal cord microvessels of ODC-OVA mice when compared to WT controls (Fig. 7C, D) suggests a specific impact of the ongoing autoimmune neuroinflammation on CD8⁺ T cell interaction with the BBB in vivo.

## CD8 T cell-mediated autoimmune neuroinflammation reduces the crawling speed of auto-Ag-specific but not other effector CD8 T cells on the inflamed BBB in vivo

As endothelial Ag presentation reduced the crawling speed of OT-I cells on pMBMECs in vitro, we next asked if such reduced crawling speeds of OT-I cells on the BBB could also be observed in ODC-OVA mice in vivo. We performed two-photon intravital microscopy (2P-IVM) of cervical spinal cord microvessels in ODC-OVA, VE-cadherin-GFP or WT C57BL/6 J mice as controls after transfer of naïve tdTomato⁺ OT-I T cells 24 h prior to LCMV-OVA infection on day 7 after LCMV-OVA infection (Fig. 7E, Supplementary Movie 7). CMFDA- labeled effector OT-I cells were systemically injected via a carotid artery catheter right before imaging. Comparing the dynamic interaction of these in vitro activated effector OT-I cells with the luminal wall of spinal cord microvessels in ODC-OVA versus control mice revealed a significantly reduced crawling speed of OT-I cells in the ODC-OVA mice (Fig. 7F). Simultaneously, we could also observe the tdTomato⁺ OT-I cells, which we had originally transferred in a naïve state into the recipient mice 24 h prior to the LCMV-OVA infection. These in vivo activated tdTomato⁺ OT-I cells were found to recirculate within the CNS vasculature, as well as infiltrate the leptomeningeal spaces in WT C57BL/6 mice or, in addition, the CNS parenchyma in ODC-OVA mice (Fig. 7E). Further underscoring a potential role of brain endothelial Ag presentation in CD8⁺ T cell interaction with the BBB, we observed a reduced crawling speed of these in vivo activated tdTomato⁺ effector OT-I cells within the spinal cord microvessels in ODC-OVA mice when

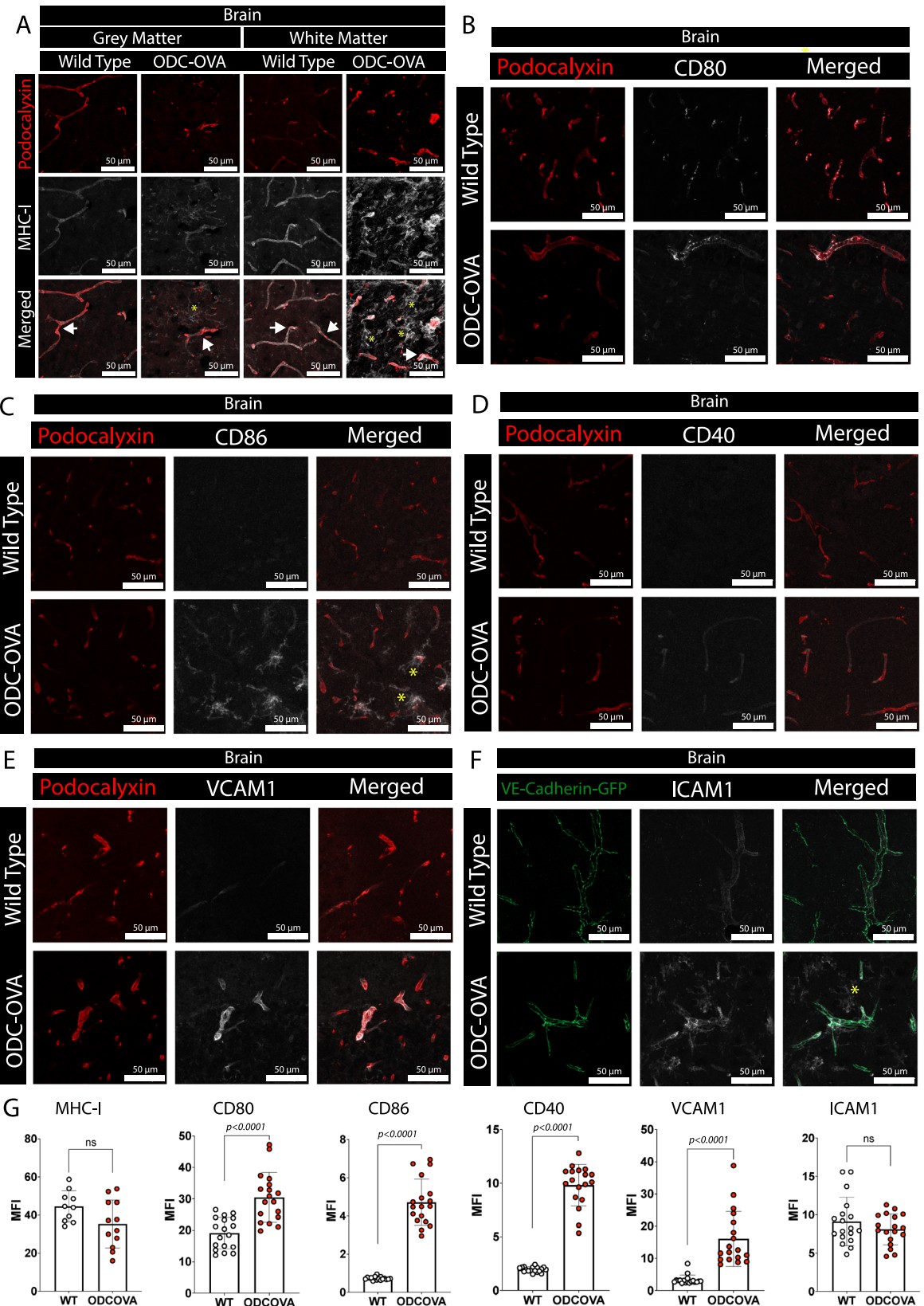

compared to WT C57BL/6 control mice (Fig. 7G). To distinguish the role of general neuroinflammation and thus increased endothelial adhesion molecule expression from that of endothelial MHC-class I restricted Ag presentation on OT-I cell crawling speed on the BBB we next investigated the interaction of CMFDA-labeled effector OT-I cells with the spinal cord microvessels in C57BL/6 J mice during the peak of

experimental autoimmune encephalomyelitis (EAE), where neuroin-flammation is induced by autoaggressive myelin-specific CD4+ T cells. The crawling speeds of OT-I cells in mice suffering from EAE were significantly faster than those observed during autoimmune neuroin-flammation in ODC-OVA mice and rather comparable to the crawling speeds observed in WT control C57BL/6 mice 7 days after LCMV-OVA

**Fig. 6 | Inflamed brain microvascular endothelial cells express MHC class I and co-stimulatory molecules.** Brains from ODC-OVA, VE-Cadherin-GFP ODC-OVA, VE-Cadherin-GFP or WT C57BL/6 J mice were harvested on day 7 after induction of autoimmune neuroinflammation and double immunofluorescence stainings were performed on 20 µm cryosections: (**A**) MHC class I (gray), and podocalyxin-positive vasculature (red). Immunostainings for the gray matter (left) and white matter (right) are depicted. Double immunofluorescence staining for CD80 (**B**), CD86 (**C**), CD40 (**D**) and VCAM-1 (**E**) and podocalyxin-positive vasculature (red) is shown. **F** Immunofluorescence staining of VE-Cadherin-GFP and VE-Cadherin-GFP ODC-OVA mouse brains for ICAM-1 (gray). White arrows show co-localization of immunostaining for MHC class I with podocalyxin-positive vessels. Yellow asterisks highlight CNS parenchymal cells staining positive for MHC class I or CD86 or ICAM-1. Scale bar = 50 µm. The data in (**A**) is representative of 5 different immunostainings on tissues from 3 different mice derived from 3 individual experiments. The data from (**B**–**F**) are representative of 3 different immunostainings on tissue from 3 different mice derived from 3 individual experiments. **G** Quantification of (**A**–**F**) represented as mean fluorescence intensity of MHC-I, CD80, CD86, CD40, VCAM-1 and ICAM-1 on the podocalyxin⁺ or VE-cadherin-GFP⁺ vessels. Data were analyzed from 3 independent experiments using two-sided unpaired parametric T-Welch's test, and shown as mean ± SD. Source data from (**G**) are provided as a Source Data file.

infection (Fig. 7F). These data thus strongly support a specific contribution of brain endothelial Ag presentation rather than general neuroinflammation and thus increased endothelial adhesion molecule expression in reducing the OT-I cell crawling speed in spinal cord microvessels of ODC-OVA mice. Also, the crawling speeds of TCR transgenic P14 and CL4 CD8⁺ T cells, whose cognate antigen (P14) is cleared or matching MHC class I (CL4) is absent in our experimental setup, were found comparable in spinal cord microvessels in ODC-OVA or WT C57BL/6 J mice at day 7 after LMCV-OVA infection (Fig. 7F), further underscoring that MHC class I-restricted recognition of cognate Ag reduces CD8⁺ T cell crawling speed on the inflamed BBB in vivo.

### Endothelial MHC-class I mediated Ag cross-presentation reduces CD8 T cell crawling speed on the inflamed BBB in vivo

To finally provide direct evidence for the impact of brain endothelial MHC class I-mediated Ag presentation on CD8⁺ T cell interaction with the BBB in vivo, we generated a mouse model with inducible BBB-specific abrogation of functional MHC-class I mediated Ag presentation. To this end we crossed ODC-OVA mice with TAP1ᶠˡ/ᶠˡ mice. The transporter associated with antigen processing 1 (TAP1) mediates the delivery of peptides across the endoplasmic reticulum membrane to MHC class I molecules. Absence of TAP1 interrupts stable assembly and intracellular transport of MHC class I molecules[23] and therefore, TAP1-deficient brain endothelial cells can no longer present antigens to CD8⁺ T cells. BBB-specific deletion of TAP1 was achieved by intravenous injection of the AAV-BR1-CAG-Cre adeno-associated viral vector[24] two weeks prior to induction of CD8⁺ T cell mediated neuroinflammation. The specific tropism of the AAV-BR1 viral vector for brain microvascular endothelial cells has been described before[24–26] and could be confirmed in ODC-OVA mice where after intravenous injection of AAV-BR1-CAG-GFP we observed expression of GFP mainly in brain microvascular endothelial cells (Supplementary Fig. 8). We observed additional expression of GFP in select neurons suggesting that the vector can be transported across the BBB as previously observed[27]. Our findings underscored the suitability of the AAV-BR1 vector to target brain microvascular endothelial cells allowing for conditional Cre-driven recombination of the floxed TAP1 allele specifically in the CNS microvascular endothelial cells of ODC-OVA TAP1ᶠˡ/ᶠˡ mice and thus establishing a BBB-specific TAP1 deficient ODC-OVA mouse model (ODC-OVA TAP1ᴮᴮᴮ⁻ᴷᴼ). OT-I cell mediated neuroinflammation (transfer of naïve OT-I cells followed by LCMV-OVA infection) was then induced in ODC-OVA TAP1ᶠˡ/ᶠˡ mice or as control in ODC-OVA TAP1ʷᵗ/ʷᵗ mice two weeks after injection of AAV-BR1-CAG-Cre, or as an additional control in ODC-OVA TAP1ᶠˡ/ᶠˡ mice two weeks after injection with AAV-BR1-CAG-GFP (Fig. 7H). Comparing the interaction of in vitro activated CMFDA-labeled effector OT-I cells with the inflamed spinal cord microvessels expressing or not TAP1 by two-photon in vivo imaging showed that OT-I cell crawling speed was significantly faster in the absence of TAP1 in the BBB when compared to their crawling speed in mice expressing TAP-1 at the BBB (Fig. 7H). These data provide direct evidence that brain endothelial MHC-class I-mediated Ag presentation leads to a reduced crawling speed of CD8⁺ T cells on the BBB in vivo.

### CD8 T cell-mediated autoimmune neuroinflammation in ODC-OVA mice is associated with mild focal BBB breakdown in vivo

Recognition of the cognate Ag on MHC class I on brain endothelial cells stopped CD8⁺ T cell crawling which resulted in CD8⁺ T cell mediated destruction of the brain endothelial monolayer in vitro. Therefore, we next asked if the reduced OT-I cell crawling speed induced by CNS endothelial Ag-cross-presentation during neuroinflammation in ODC-OVA mice would also result in BBB breakdown in vivo. To this end we performed immunostainings for IgG and cleaved caspase 3 in brain sections from VE-Cadherin-GFP ODC-OVA mice and VE-Cadherin-GFP control mice 7 days after LCMV-OVA infection and prior naïve OT-I transfer (Fig. 7I and Supplementary Fig. 6). As a positive control, immunostainings for IgG and cleaved caspase 3 were also performed on in brain sections from VE-cadherin-GFP C57BL/6 J mice derived from mice at the peak of EAE where BBB breakdown is documented (Fig. 7I and Supplementary Fig. 6). We did not detect any extravascular IgG staining in VE-Cadherin-GFP control mice, indicating an intact BBB. On the other hand, we observed sporadic focal BBB lesions in the VE-Cadherin-GFP ODC-OVA mice shown by the extravascular detection of IgG often found associated with perivascular immune cell accumulation, which is comparable to the characteristic extravascular IgG staining observed around the leaky BBB in EAE (Fig. 7I). Immunostaining for cleaved caspase 3 was observed in tight association with VE-cadherin-GFP⁺ vessels in ODC-OVA mice (Supplementary Fig. 6). The final assignment of these vessel associated apoptotic cells as endothelial cells, pericytes or even perivascular immune cells was however difficult due proximity of these cell types to each other (Supplementary Fig. 6).

### Cytotoxic activity of CD8 T cells is not required for their migration across the BBB

To understand if BBB breakdown mediated by release of granzyme B by OT-I cells following Ag-recognition on the BBB is required for their CNS entry, we induced the CD8⁺ T cell driven autoimmune neuroinflammation in VE-cadherin-GFP ODC-OVA mice by transferring either naïve OT-I or naïve OT-I GrB⁻/⁻ cells one day prior to infection with LCMV-OVA. We did not observe any significant difference in the disease course or overall disease severity between the two groups (Supplementary Fig. 7A, B). In addition, no qualitative differences were observed in the VE-Cadherin-GFP ODC-OVA mice suffering from OT-I or OT-I GrB⁻/⁻ cell induced autoimmune neuroinflammation with respect to CNS infiltration with CD45⁺ immune cells or deposition of extravascular IgG at day 7 after LCMV-OVA infection (Supplementary Fig. 7C, D). Thus, Ag-recognition by CD8⁺ T cell inducing CD8⁺ T cell mediated, Granzyme B-dependent BBB breakdown is not essential for immune cell infiltration and focal BBB breakdown in the ODC-OVA model of autoimmune neuroinflammation.

### Leptomeningeal and perivascular phagocytes efficiently take up ovalbumin in vivo

Comparing our in vitro and in vivo observations we speculated that cross-presentation of OVA on BBB endothelium during neuroinflammation in ODC-OVA mice in vivo may be very low and, while allowing for reducing OT-I crawling speed, but not sufficient to induce

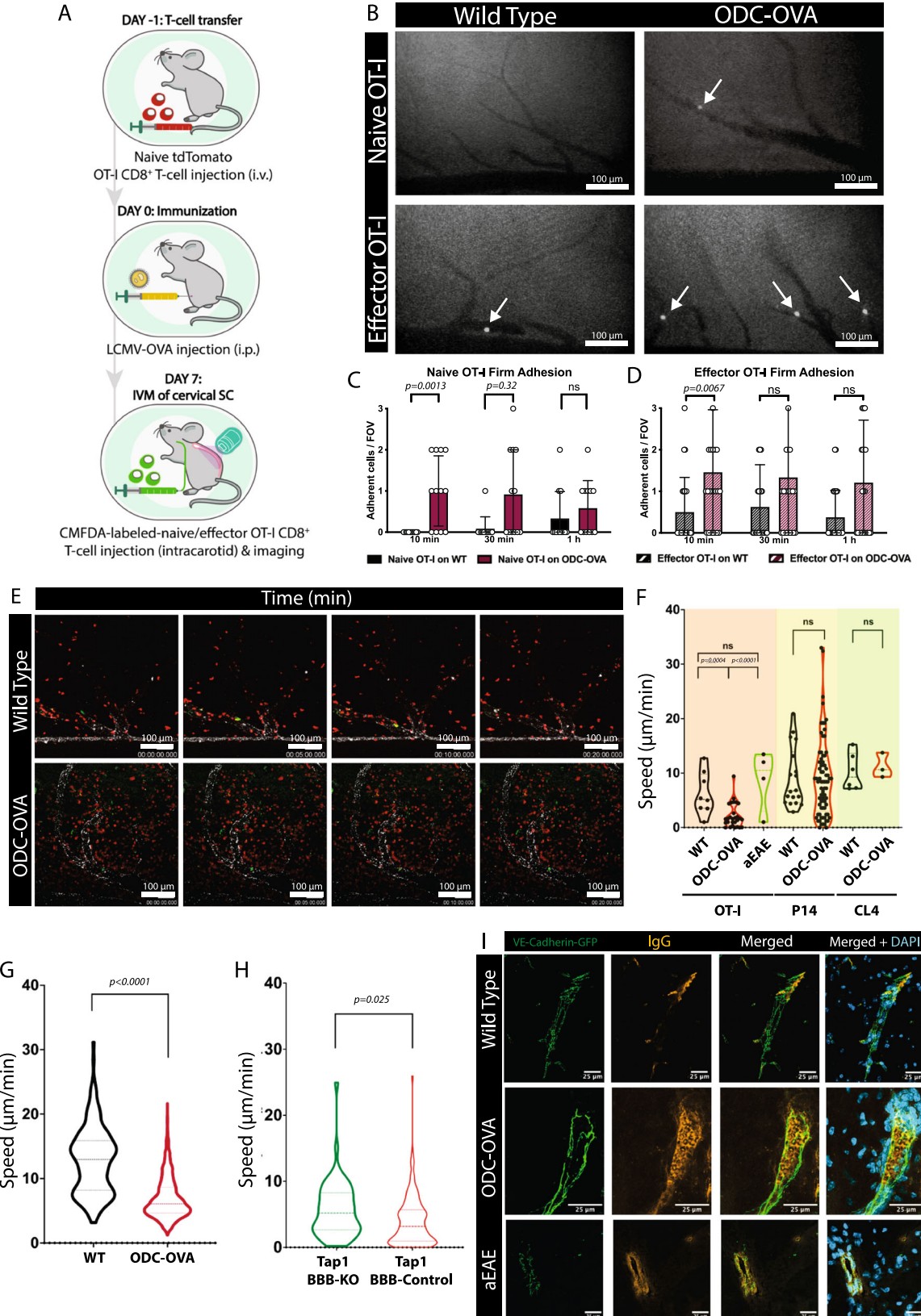

OT-I mediated breakdown of the BBB. Therefore, we finally asked if and to what extent CNS microvascular endothelial cells could take up ovalbumin potentially released from oligodendrocytes during neuroinflammation from their abluminal side in vivo. To this end we injected OVA-AF488 via the cisterna magna into naïve C57BL/6 J (Fig. 8A) or ODC-OVA mice on day 7 after LCMV-OVA infection (Fig. 8B)

during cervical spinal cord 2P-IVM imaging. Interestingly, independent of the presence or absence of neuroinflammation, OVA-AF488 was efficiently taken up by leptomeningeal and perivascular phagocytes, which are likely CNS-associated macrophages (CAMs) based on their localization along the pia mater and the vascular tree[28](Fig. 8A, B). At the same time, we could not visualize the uptake of OVA-AF488 in CNS

**Fig. 7 | CD8⁺ T cells show an increased arrest coefficient as well as lower general motility on the inflamed BBB in vivo. A** Experimental setup for the induction of CD8⁺ T cell driven neuroinflammation in the ODC-OVA mouse and its imaging. **B** Representative images of epifluorescence-IVM imaging of CMFDA-labeled-naïve and -effector OT-I cells adhering in inflamed cervical spinal cord microvessels of WT and ODC-OVA mice 60 min post-infusion. White arrows show arrested OT-I cells in cervical spinal cord microvasculature. Number of naïve (**C**) and effector (**D**) OT-I cells adhered in inflamed cervical spinal cord microvessels of WT and ODC-OVA mice with neuroinflammation at 10, 30 and 60 min after infusion of OT-I cells. Data were pooled from 3 mice/genotype for naïve- and 4 mice/genotype for effector OT-I cells, analyzed using two-tailed non-parametric Mann-Whitney U test, and shown as ± SEM. **E** Representative images over time of two-photon imaging of tdTomato⁺ in vivo activated (red) and CMFDA-labeled in vitro activated OT-I cells (green) adhered in inflamed cervical spinal cord microvessels (white) of WT or ODC-OVA mice 60 min post-infusion on day 7 after viral infection. Pictures are depicted as maximum intensity projection of 100 µm thick Z-stacks. See Supplementary Movie 7. **F** Violin plots for crawling speeds (µm/min) of CMFDA-labeled effector CD8⁺ T cells generated from either OT-I, P14 or CL4 TCR-transgenic mice as observed in WT C57BL/6 J or ODC-OVA mice at day 7 after induction of auto-immune neuroinflammation or during EAE. Prior to the intravital imaging, in vitro activated CD8⁺ T cells were injected into the circulation. Data were pooled from 3 mice/condition and each dot represent one CD8⁺ T cell track. Data were analyzed using two-sided unpaired parametric t test. **G** Crawling speed of recirculating in vivo activated tdTomato⁺ OT-I cells injected 24 h prior to LCMV-OVA infection in WT C57BL/6 J and ODC-OVA mice at day 7 after LCMV-OVA infection. Violin plots summarize data from 196 OT-I cells in WT mice and 991 OT-I cells in ODC-OVA mice. Data were analyzed using two-sided un-paired parametric t test. **H** Violin plots for crawling speed (µm/min) of in vitro activated CMFDA-labeled OT-I cells in the cervical spinal cord microvasculature lacking expression of TAP1 (TAP1 BBB-KO) or not (TAP1-BBB-Control). TAP1- BBB KO mice (green): BBB-specific deletion of TAP1 in ODC-OVA// TAP1^floxed/floxed mice by injection of the AAV-BR1-CAG-Cre viral vector two weeks prior to induction of autoimmune neuroinflammation. TAP1-BBB-Control (red): ODC-OVA//TAP1^WT/WT mice injected with AAV-BR1-CAG-Cre and ODC-OVA//TAP1^floxed/floxed mice injected with AAV-BR1-CAG-GFP. OT I cells were injected prior to intravital imaging. Data are pooled from imaging of 3 TAP1 BBB-KO mice and 6 TAP1-BBB-Control mice and analyzed using two-sided unpaired parametric t test. **I** Brains from VE-Cadherin-GFP ODC-OVA or VE-Cadherin-GFP control mice were harvested on day 7 after LCMV-OVA infection. Immunofluorescence staining for IgG and nuclei (DAPI). The endogenous GFP signal marks vascular adherens junctions. Data are representative of 5 individual stainings from 3 different mice derived from 3 individual experiments. Source data from (**C, D, F, G** and **H**) are provided as a Source Data file.

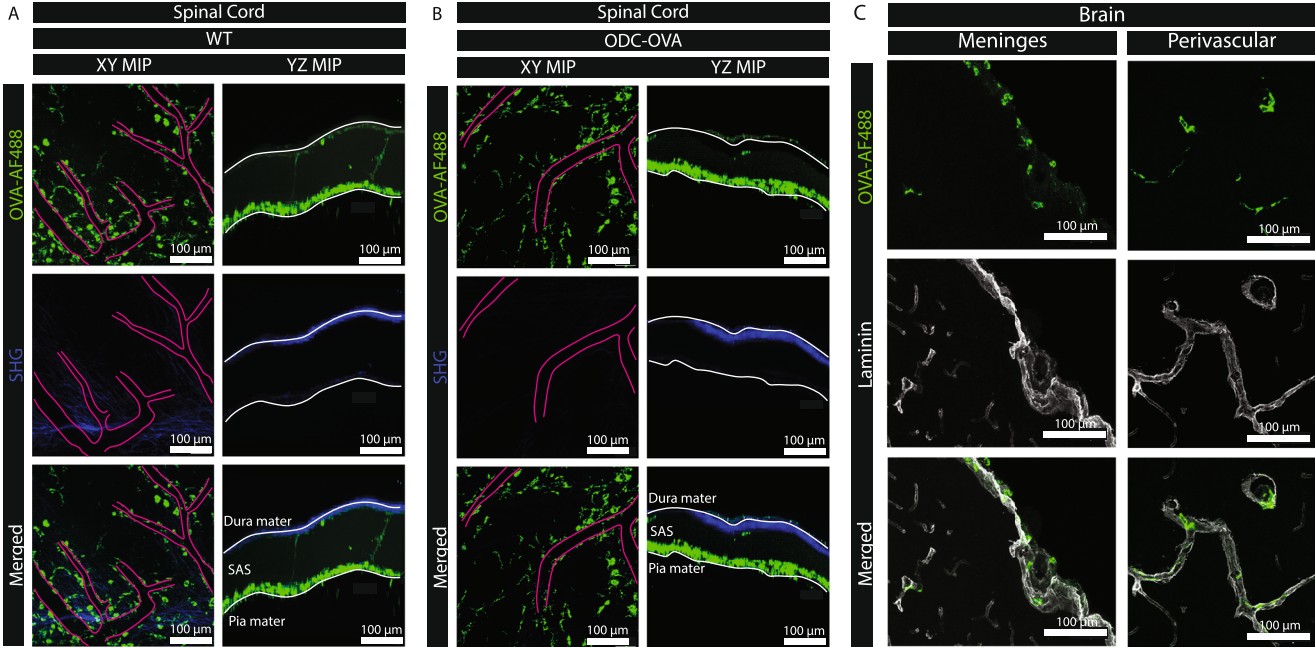

**Fig. 8 | Leptomeningeal and perivascular border-associated macrophages efficiently take up CSF ovalbumin in vivo.** 2P-IVM imaging of the cervical spinal cord leptomeninges of naïve WT C57BL/6 J mice (**A**) or ODC-OVA mice on day 7 after induction of autoimmune neuroinflammation (**B**) 2 h after injection of 2 µg of OVA-AF488 into the cisterna magna (rate 1 µL/min) is shown. Images show OVA-AF488 (green) and second harmonic generation (blue) generated by collagen type I in the dura mater and the extracellular matrix in the subpial space. Scale bar = 100 µm. **C** Immunofluorescence staining for laminin (gray) on 20 µm thick cryo-section from brains isolated from healthy WT C57BL/6 J mice 4 h after cisterna magna injection of 2 µg OVA-AF488 (rate 1 µL/min) (green) is shown. Left panel shows cellular uptake of OVA-AF488 in the leptomeninges and the right panel in perivascular spaces. Representative data from 2 mice/genotype.

endothelial cells neither in the presence or in the absence of neu-roinflammation by live cell imaging. Subsequent confocal image ana-lysis of brain and spinal cord sections of the OVA-AF488 injected mice that were co-stained for laminin confirmed the perivascular and lep-tomeningeal localization of the OVA-AF488 tracer, as well as the uptake of OVA-AF488 in CAMs, but did not allow for detection of OVA-AF488 based on the fluorescent signal in CNS microvascular endo-thelial cells (Fig. 8C). Because we observed that the CNS endothelial MHC-class I-mediated Ag presentation in vivo reduced the crawling speed of Ag-specific but not Ag-non-specific CD8⁺ T cells, we conclude that CNS endothelial cells can take up, process and present OVA-AF488 but the level of uptake in vivo stays below the detection limit of fluorescence microscopy.

Taking all observations together our data suggest that in the ODC-OVA model OVA released during autoimmune neuroinflammation is taken up by CAMs and to a lower degree by the BBB endothelial cells. In accordance to our in vitro observations, where high levels of Ag pre-sentation on the BBB stopped CD8⁺ T cell migration the apparently lower levels of Ag presentation on the BBB in the ODV-OVA model in vivo do suffice to reduce the crawling speed of Ag-specific CD8⁺ T cells on the luminal side of the BBB. Thus Ag presentation at the level of the BBB does not promote CNS invasion of CD8⁺ T cells. Rather, in correlation to the level of luminal MHC class I-restricted Ag presenta-tion, recognition of their cognate Ag on the BBB will prohibit CD8⁺ T cell entry into the CNS and eventually lead to focal BBB breakdown as observed in MS.

## Discussion

In the present study, we show that BBB endothelial cells can process exogenous protein antigens from both their abluminal and luminal side and present it on their luminal side in an MHC class I-dependent manner to CD8[+] T cells in vitro. Effective Ag-dependent activation of naïve T cells is a process usually involving two signals. The first Ag-dependent signal is provided by the engagement of the T cell receptor with its cognate Ag presented on the MHC molecule on the APC. The second signal is Ag-independent and mediated by co-stimulatory molecules such as CD80 and CD86 on the APC, engaging CD28 on the T cell. The need for a second signal in T cell activation is thought to ensure the maintenance of self-tolerance.

Here we observed that inflamed pMBMECs despite their very low expression of CD80 and CD86, supported priming and activation of naïve CD8[+] T cells in an MHC class I-dependent manner leading to CD8[+] T cell proliferation and full effector cell differentiation as shown by the induction of brain endothelial apoptosis and ultimately rupture of the pMBMEC monolayer. Reports on the endothelial expression of CD80 and CD86 have been controversial with some studies showing the expression of both co-stimulatory molecules while others were unable to detect mRNA or protein expression of these molecules in endothelial cells (summarized in[17]. However, human brain endothelial cells were previously shown to support allogeneic activation and proliferation of naïve CD4[+] and CD8[+] T cells in the absence of CD80 and CD86 expression suggesting the existence of mechanisms allowing to bypass this co-stimulatory pathway in T cell activation. Moreover, some T cell responses can be induced in mice lacking expression of CD80/CD86 or CD28[29]. In the case of endothelial cells, molecules such as CD40 and ICOS-L[30] or adhesion molecules such as ICAM-1 and VCAM-1 were shown to provide co-stimulatory signals for T cell activation (summarized in[17]. Thus, on cytokine stimulated pMBMECs low expression of CD80 and CD86 may be compensated by the expression of other co-stimulatory molecules including ICAM-1 and VCAM-1[31]. Alternatively, endothelial MHC class I-peptide complexes may suffice to activate naïve OT I CD8[+] T cells as previously shown in studies aiming to delineate the minimal requirements for naïve CD8[+] T cell activation by using APC-free systems[32,33]. These results suggested that a potent TCR signal can bypass the requirement for co-stimulation. Considering that OT-I CD8[+] T cells recognize the ovalbumin derived peptide SIINFEKL bound to H2K[b] with high affinity[34], we, therefore, chose an experimental setup where we pulsed pMBMECs with very low concentrations of SIINFEKL (0.5 nM or lower) to avoid high Ag-density on pMBMECs[32]. Methodology allowing to specifically quantify the peptide-MHC class I complexes on the surface of pMBMECs would allow measuring the Ag density on pMBMECs and the requirement of co-stimulation for naïve CD8[+] T cell activation by pMBMECs.

An additional observation we made is that inflamed pMBMECs possess the entire machinery allowing for cross-presentation of Ag[35] as they can take up, process and present exogenously provided ovalbumin on MHC class I molecules on their luminal surface leading to naïve OT-I activation. Our observations are in accordance with previous studies showing that brain endothelial cell cross-presentation of parasite Ags during experimental cerebral malaria (CM) induces CD8[+] T cell mediated BBB dysfunction[36]. In a mouse model of CM brain endothelial cell-specific ablation of H-2Kb or H-2Db was found to reduce CD8[+] T cell interaction with the brain vasculature which prevented BBB breakdown and death of the animals[37]. In CM, brain endothelial cells take up parasite Ags from their luminal side to subsequently cross-present these Ags again on their luminal surface to circulating CD8[+] T cells. Our present study shows, however, that inflamed brain microvascular endothelial cells can also process exogenous protein Ags provided from their abluminal side and present it on MHC class I molecules to naïve CD8[+] T cells on their luminal side leading to their activation and proliferation and differentiation to effector CD8[+] T cells inducing brain endothelial apoptosis. These observations suggest that during neuroinflammation activated BBB endothelial cells may take up CNS-derived Ags from their abluminal side and subsequently cross-present these Ags on their luminal surface in MHC class I to circulating CD8[+] T cells. It was previously proposed that recognition of their cognate antigen on MHC class I would facilitate CD8[+] T cell entry into the CNS across the BBB[11].

T cells circulating in the bloodstream are, however, exposed to shear forces and thus recognition of their cognate Ag on the BBB would require the productive engagement of the TCR with the endothelial peptide-MHC class I complex under physiological flow. In vitro and in vivo live cell imaging has provided evidence that T cell extravasation across the inflamed BBB is a multi-step process regulated by the sequential interaction of adhesion and signaling molecules between the endothelial cells and the T cells (summarized in[8]). We have previously shown that activated CD8[+] T cells can cross pMBMECs under physiological flow in vitro in an antigen-independent manner by LFA-1 mediated shear-resistant arrest on endothelial ICAM-1 and ICAM-2 followed by their polarization and crawling and finally their G-protein receptor (GPCR)-dependent diapedesis across the pMBMEC monolayer preferentially via transcellular pores[9]. Investigating the interaction of naïve and effector CD8[+] T cells with the inflamed pMBMECs under physiological flow we here found that effector OT-I cells arrested on the stimulated pMBMECs with significantly higher efficiency than naïve cells. Endothelial Ag presentation did not affect the shear-resistant arrest of naïve nor the effector OT-I cells suggesting that TCR engagement of peptide-MHC molecules on the endothelial surface does not occur under physiological flow. While following their shear-resistant arrest the majority of naïve but not effector OT-I cells readily detached from the pMBMEC monolayers, detachment of naïve OT-I cells was prohibited in presence of their cognate Ag and MHC class I. This observation underscores that during transient arrest on pMBMECs naïve CD8[+] T cells can engage with their TCR endothelial peptide-MHC class I complexes leading to increased avidity of their binding to the brain endothelium probably via inside-out activation of their integrins allowing for high-affinity binding of their endothelial ligands[38]. Although endothelial Ag presentation allowed for adhesion strengthening of naïve CD8[+] T cells to the pMBMEC monolayer under flow, it abrogated T cell crawling on the pMBMECs in line with previous observations showing that engagement of the TCR with the MHC-peptide complexes induces a stop signal for T cells and hinders their crawling[39]. Accordingly, we observed that Ag presentation by inflamed pMBMECs also induced such a stop signal for effector OT-I cells by abrogating their crawling and subsequent diapedesis across the pMBMEC monolayer. Thus, provided their productive initial engagement on brain endothelial cells under physiological shear, naïve and effector CD8[+] T cells can in a subsequent step recognize their cognate Ag in an MHC class I-dependent manner on brain endothelial cells. It has previously been shown that highly motile T cells can stop when encountering only a few MHC-antigen complexes on the surface of another cell[39–42]. In this context, the competition of TCR stop signals with chemokine-mediated go signals was shown to modulate the duration of immunological synapse formation involved in shaping the immune response towards tolerance or activation[39,40,42]. Our observations are in line with this concept that recognition of their cognate Ag in an MHC class I-dependent manner on the BBB provides a stop signal for CD8[+] T cells that is stronger than the go signals displayed by adhesion and signaling molecules. Unlike previously proposed, Ag-recognition on brain endothelial cells does thus not facilitate CD8[+] T cell diapedesis across the BBB. It rather triggers their arrest and increases their probing behavior on the luminal surface of the BBB potentially stabilizing the immunological synapse. In our study, these arrested and probing CD8[+] T cells induced endothelial apoptosis and breakdown of the pMBMEC monolayer under physiological flow in vitro. Our observations, therefore, suggest that luminal MHC class I-restricted Ag presentation by the BBB rather than facilitating CD8[+] T

cell migration across the BBB will arrest CD8[+] T cells on the luminal side of the BBB eventually triggering focal BBB breakdown as observed in MS.

In MS, demyelination, oligodendrocyte death and axonal loss are the major consequences of chronic neuroinflammation. As our in vitro observations showed that pMBMECs were able to cross-present Ag provided from their abluminal side on their luminal surface to CD8[+] T cells we also aimed to understand if cross-presentation of CNS antigens at the luminal BBB can be observed in vivo. We made use of ODC-OVA mice expressing ovalbumin as a sequestered CNS model antigen in the cytoplasm of oligodendrocytes[18]. Upon infection with LCMV-OVA, peripherally activated OT-I cells enter the CNS parenchyma of ODC-OVA but not WT control mice leading to neuroinflammation, oligodendrocyte damage and clinical disease[20,43]. As infection of endothelial cells with the LCMV strain used in our present study has not been observed[44] and the virus is readily cleared from the mouse[21], we argued that brain endothelial MHC class I-dependent cross-presentation of oligodendrocyte derived OVA should lead to reduced motility of OT-I cells in CNS microvessels in ODC-OVA mice, when compared to control mice in vivo. Supporting this notion, we observed increased arrest of both, naïve and effector OT-I cells in ODC-OVA mice in spinal cord microvessels. Recapitulating our in vitro observations, we observed a significant reduction in the crawling speed of effector OT-I cell in the spinal cord microvessels of ODC-OVA mice when compared to WT control mice. Combining different experimental in vivo live cell imaging approaches, in which we either studied the interaction of effector CD8[+] T cells whose cognate antigen or a matching MHC class I was absent in our experimental setup or where we chose a model of neuroinflammation lacking endogenous OVA expression, we here show that reduced crawling speed of effector CD8[+] T cells on the inflamed BBB could only be observed with OT-I cells in the context of neuroinflammation in ODC-OVA mice. At the same time expression levels of endothelial ICAM-1, an adhesion molecule previously shown to mediate T cell crawling at the BBB[45], were comparable in WT control and ODC-OVA mice underscoring that the reduction of T cell crawling speed required expression of the cognate CD8[+] T cell Ag in the CNS and could not be explained by upregulated expression of ICAM-1 on the BBB. Indeed, BBB-specific abrogation of functional MHC class I-dependent Ag presentation by AAV-BR1-Cre-mediated deletion of TAP1 directly confirmed that the reduced crawling speed of effector CD8[+] T cells on the inflamed BBB is due to MHC-class I-restricted Ag-recognition by the CD8[+] T cells.

To obtain a better insight into CNS endothelial Ag-uptake from the CNS side in vivo we explored the uptake of cisterna magna applied OVA-A488 and observed that leptomeningeal and perivascular CNS-associated macrophages (CAMs)[28] are very efficient in taking up fluorescently tagged OVA, while uptake of OVA-A488 by CNS endothelial cells remained below the detection level of our fluorescent imaging. The important role of CAMs in maintaining CNS homeostasis and enforcing the brain borders has only recently been recognized since transcriptome profiling has allowed making them clearly distinguishable from microglial cells residing in the CNS parenchyma[46]. The precise role of CAMs in neuroinflammation remains to be shown. They have been suggested to regulate CSF flow[47] in addition to their essential role in presenting antigen to encephalitogenic T cells in the leptomeningeal space to initiate neuroinflammation[48]. In the context of our present study, it is therefore tempting to speculate that CAMs display immunomodulatory functions by phagocytosing excess CSF-derived Ags which may limit Ag-cross presentation by CNS endothelial cells. Limited uptake of CNS antigens and their cross-presentation by the BBB however does still suffice to reduce CD8[+] T cell crawling speed within CNS microvessels but will not readily lead to massive CD8[+] T cell mediated BBB breakdown. Thus, unlike previously proposed[11] rather than facilitating CD8[+] T cell entry into the CNS, MHC class I-restricted Ag presentation by the BBB endothelial cells arrests CD8[+] T cells at the luminal BBB. Depending on the levels of MHC-class I-dependent Ag presentation this interaction will lead to brain endothelial apoptosis as observed by us in vitro or to reduced CD8[+] T cell crawling speeds as observed in the ODC-OVA model in vivo. In the latter context the efficient CNS-derived antigen uptake by CAMs may reduce the availability of Ag-MHC-class I complexes on the surface of the BBB to a level that is sufficient to reduce CD8[+] T cell crawling speed but not necessarily to trigger CD8[+] T cell mediated BBB breakdown. This hypothesis is in line with a previous study where in a transgenic mouse model the targeted expression of a neo-antigen in brain endothelial cells did result in hemorrhagic BBB breakdown upon transfer of neo-antigen specific effector CD8[+] T cells[49]. This underscores that the density of peptide-MHC class I complexes on the luminal BBB may be decisive for the degree of damage that can be induced by circulating Ag-specific CD8[+] T cells. That in turn depends on the expression levels of endothelial MHC class I and co-stimulatory molecules, which are as our study shows upregulated during neuroinflammation, but also the amount and the nature of CNS tissue damage as well as the involvement of CAMs and potentially microglial cells, which we hypothesize will influence the availability of Ag to be taken up and processed by brain endothelial cells.

Importantly, we found that the initial adhesive interactions of CD8[+] T cells with brain microvascular endothelial cells under shear are independent of Ag recognition. Thus, only those CD8[+] T cells that can initiate productive adhesive interactions with the BBB will be able to subsequently recognize their cognate Ag on the BBB. In the absence of neuroinflammation adhesion molecule expression on BBB endothelium is very low and T cell interaction is restricted to low numbers of activated T cells[8] suggesting that Ag recognition by CD8[+] T cells on the BBB rarely occurs under these conditions. Increased adhesion molecule expression during neuroinflammatory diseases allows for increased adhesive interactions mediated by adhesion molecules like α4β1-integrin (VLA-4) which also mediates CD8[+] T cell entry into the CNS[50]. It is tempting to speculate that the success of therapeutic targeting of this molecule for the treatment of MS not only relies on inhibition of T cell migration across the BBB but in parallel on prohibiting Ag-specific CD8[+] T cell interactions with the brain endothelium and thus CD8[+] T cell mediated BBB breakdown. Future studies on the role of CAMs and how the nature of the respective Ag, the density of Ag presentation on MHC class I molecules on brain endothelial cells affects the multi-step extravasation of CD8[+] T cells across the BBB and BBB integrity will be relevant to improve our understanding of the pathology underlying MS and other neuroinflammatory disorders.

## Methods
### Mice
Male and female mice from C57BL/6 J strain between 8–12 weeks of age were used in this study. Wildtype C57BL/6 J mice were obtained from Janvier (Genest Saint Isle, France). β2-microglobulin[−/−] (B6.129P2-B2mtm1Jae) mice lacking the functional expression of MHC class I[51] were provided by Swiss Immunological Mouse Repository (SWImMR; Zürich). Lifeact-GFP mice (C57BL/6-Tg(CAG-EGFP)#Rows) for studying actin dynamics have been described before[52]. OT-I mice (C57BL/6J-Tg(Tcra/Tcrb)1000Mjb) contain a transgenic TCR, recognizing OVA residues 257-264 (SIINFEKL) in the context of H2-K[b34]. OT-I Rag-1[−/−] C57BL/6 J mice were generated by crossing OT-I mice with Rag-1[−/−] mice (Rag1[tm1Mom]). OT-I tdTomato mice expressing tdTomato reporter ubiquitously were generated by crossing OT-I mice with Ai14 tdTomato reporter mice (Gt(ROSA)26Sortm14(CAG-tdTomato)Hze), in which the Stop-cassette had previously been deleted by breeding with ZP3-Cre transgenic (Tg(Zp3-cre)93Knw) mice. Mice expressing a VE-Cadherin C-terminal EGFP-fusion protein from the endogenous locus (Cdh5tm9Dvst) have been described before[22,31,53]. NG2-DsRed (Tg(Cspg4-DsRed.T1)1Akik mice[54] were provided by Ralf Adams (MPI, Münster). OT-I Granzyme B[−/−] mice were generated by crossing OT-I

mice with Granzyme B$^{-/-}$(Gzmbtm1Ley) mice[55] kindly provided by Charaf Benarafa (IVI, Mittelhäusern, Switzerland). ODC-OVA mice have been described before[18]. VE-cadherin-GFP ODC-OVA mice were generated by crossing ODC-OVA mice with the VE-cadherin-GFP knock-in mice[22]. TAP1$^{floxed/floxed}$ were generated from the conditional ready Tap1$^{tm2a(EUCOMM)Hmgu}$ mice from the European Conditional Mouse Mutagenesis Program (EUCOMM) by flippase-mediated excision of the LacZ and Neomycine resistance sequences. Double homozygous ODC-OVA TAP1$^{floxed/floxed}$ mice were generated by serial crossings of ODC-OVA mice with TAP1$^{floxed/floxed}$ mice. TCR transgenic P14 mice (B6.Cg-Tcra$^{tm1Mom}$Tg(TcrLCMV)327Sdz) mice are transgenic for a TCR, which recognizes the LCMV gp33–41 (KAVYNFATM) in an H2-D$^b$ restricted manner[56]. P14 mice on a Rag1$^{-/-}$ background were kindly provided by Philippe Krebs (Institute of Pathology, University of Bern,Switzerland).

All gene targeted mice in the C57BL/6 background were backcrossed to the C57BL/6 J background for at least 10 generations. All mice were housed in individually ventilated cages under specific pathogen-free conditions at 22 °C with a 13:11 h light cycle and with free access to chow and water. Animal procedures were approved by the Veterinary Office of the Canton Bern (permit no. BE31/17 and BE55/20) and are in line with institutional and standard protocols for the care and use of laboratory animals in Switzerland.

### In vitro blood-brain barrier model
Primary mouse brain microvascular endothelial cells (pMBMECs) were isolated from 7 to 10 weeks old β$_2$-microglobulin$^{-/-}$ and their C57BL/6 J WT littermates or from VE-Cadherin GFP knock in C57BL/6 J mice and cultured exactly as previously described[45,57]. Barrier properties of pMBMECs and their suitability as in vitro model for the blood-brain barrier (BBB) as well as for studying immune cell extravasation across the BBB under physiological flow has been described in depth before[45,57,31,58]. Intact monolayers were stimulated with recombinant mouse TNF-α (5 ng/mL, Vitaris AG, Baar, Switzerland) and recombinant mouse IFN-γ (100 U/mL, PreproTech EC Ltd., London, UK) 24 h prior to the assays.

### CD8$^+$ T cell isolation and differentiation
**Naïve CD8$^+$ T cell isolation.** Peripheral lymph nodes and spleens from Rag-1$^{-/-}$ OT-I, OT-I or tdTomato OT-I C57BL/6 J mice were harvested and single cell suspensions were obtained by homogenization and filtration through a sterile 100 μm nylon mesh. A second filtration was applied after erythrocyte lysis (0.83% NH$_4$Cl, Tris-HCl). OT-I cells were isolated with magnetic CD8$^+$ T cell selection beads (EasySep, STEMCELL Technologies). The purity of the CD8$^+$ T cells was assessed by flow cytometry and was >98.5% in each experiment.

**In vitro activation of naïve CD8$^+$ T cells.** OT-I CD8$^+$ T cells were isolated and activated from OT-I, tdTomato OT-I, Perforin$^{-/-}$ OT-I or Granzyme B$^{-/-}$ OT-I mice exactly as described before[9,59]. Activated T cells were cultured in IL-2 containing media for 3 days post-activation.

**In vivo activation of naïve CD8$^+$ T cells.** $2 \times 10^5$ naïve td-Tomato OT-I cells were intravenously (i.v.) injected into WT C57BL/6 J mice 24 h prior to intraperitoneal (i.p.) infection with $10^5$ plaque-forming unit (PFU), OVA expressing lymphocytic choriomeningitis virus (LCMV-OVA)[21]. Spleens of recipient mice were collected 8 days after viral infection and CD8$^+$ T cells were purified by magnetic bead selection (EasySep, STEMCELL Technologies). Fluorescence-activated cell sorting is used for the separation of tdTomato OT-I CD8$^+$ cells from the CD8$^+$ T cells of recipient WT mice.

**In vitro activation of naïve P14 CD8$^+$ T cells.** Freshly isolated naïve CD8$^+$ T cells from P14 mice were incubated at a concentration of $10^6$ cells/mL for 48 h in IL-2 containing media with anti-CD3 (1ug/mL; *bioxcell; 145-2c11*) anti-CD28 (1ug/mL; BioLegend; 102102).

### In vitro activation of naïve CL4 cells.
HA-specific activated CD8$^+$ T cells were generated as described[60]. Briefly, purified naive CD8$^+$ T cells from CL4-TCR mice were cultured for 5 days with IL-2 (1 ng/ml), IL-12 (20 ng/ml), HA$_{512-520}$ peptide (1 μg/ml) and irradiated splenocytes. After culture, living cells were collected by Ficoll density separation and routinely contained >95% of CD8$^+$Vβ8$^+$ cells and >70% IFNγ/TNFα producing CD8$^+$ T cells as assessed by flow cytometry.

### Bone marrow derived dendritic cell isolation and culture
Bone marrow derived dendritic cells (BMDCs) were isolated from WT C57BL/6 J mice as described before[61]. In brief, following removal of the femurs and tibiae, the tips of these bones were cut open and the BM cells were isolated by centrifugation (4 min, 1500 g). Collected BM cells were cultured untouched in 20 ml cultures in petri dishes containing 18 ml restimulation medium (RPMI-1640 supplemented with 10% FBS (Thermo Fisher Scientific), 10 U/ml penicillin-streptomycin, 2 mM L-glutamine, 1% non-essential amino acids, 1 mM sodium pyruvate, and 0.05 mM β-mercaptoethanol (Grogg Chemie AG) and 2 ml Flt-3L-containing supernatant [produced from SP2/0 transfected cell line secreting mouse recombinant Flt-3L[62] for 7 days at 37 °C until activation with 100 ng/mL LPS (Sigma-Aldrich) 18 h prior to the assays.

### Immunofluorescence staining
Confluent pMBMEC monolayers were stained as described before[45]. Primary and secondary antibodies are listed in Supplementary Table 1. The assessment of apoptosis in pMBMECs was performed by immunofluorescence staining with the Image-iT™ LIVE Red Poly Caspases Detection Kit (Thermo Fisher Scientific, Massachusetts, USA) according to the manufacturer's protocol. As a positive control, apoptosis of pMBMECs was induced by incubation with staurosporine (1 μM; Abcam, Cambridge, UK). Images were acquired using a Nikon Eclipse E600 microscope connected to a Nikon Digital Camera DXM1200F with the Nikon NIS-Elements BR3.10 software (Nikon, Egg, Switzerland) or an LSM 800 (Carl Zeiss, Oberkochen, Germany) confocal microscope. Images were processed by using ImageJ software (ImageJ software, National Institute of Health, Bethesda, USA) and mounted in Adobe Illustrator software.

For immunofluorescence of tissue sections, deeply anesthetized mice were intracardially perfused with 4% PFA prior to organ harvest. Brains and spinal cords were postfixed in 4% PFA o/n at 4 °C and incubated in 30% sucrose solution for 72 h at 4 °C. Tissues were later embedded in OCT prior to freezing on dry ice and cryopreservation. 20 μm thick cryosections were made and used for immunostainings. Tissue sections were rehydrated with PBS for 10 min and incubated with blocking buffer (5% skimmed milk, 0.3% Triton X-100, 0.04% NaN$_3$ in TBS (pH = 7.4) for 20 min at room temperature prior to incubation with the primary antibody. Incubations with primary and secondary antibodies were performed at room temperature for 1 h each with extensive washing steps using PBS in between. Primary and secondary antibodies are listed in Supplementary Table 1.

### Quantitative real-time PCR (qRT-PCR)
Sample RNA extraction was performed from freshly isolated C57BL/6 J WT pMBEMCs following the manufacturers instruction of the High Pure RNA Isolation Kit (Hoffman-La Roche). cDNA was obtained from total isolated RNA of each sample with the SuperScript III First-Strand Synthesis System (Invitrogen, Carlsbad, CA, USA). mRNA expression was analyzed exactly as described before[63]. Beta actin mRNA levels were used as endogenous control. The sequences of the primers used for each gene are presented in Supplementary Table 2.

### Flow cytometry
Cell surface molecules of T cells and DCs were stained with appropriate combinations of fluorophore-conjugated mAbs at saturating concentrations on ice in the dark for 30 min. Intracellular staining of T cells

was performed exactly as described before[61]. The antibodies used and the working concentrations are listed in Supplementary Table 1. The data was acquired using an Attune NxT flow cytometer (ThermoFisher Scientific, Massachusetts, USA) and analyzed using FlowJo 10 software.

## Co-culture assays of naïve CD8+ T cells with pMBMECs/BMDCs

The activation of naïve OT-I cells was assessed by flow cytometry. TNF-α/IFN-γ-stimulated WT or B2M$^{-/-}$ pMBMECs were pulsed on the luminal side in 96-well format with either SIINFEKL (POV-3659-Peptides International, Kentucky, USA) or the (VSV) nucleoprotein peptide (RGYVYQGL, BACHEM, Bubendorf, Switzerland) at 0.1 ng/mL for 30 min at 37 °C or left unpulsed. The VSV-peptide binds to H-2K$^b$ but is not recognized by the TCR of the OT-I cells[12] and was therefore used as a control.

Abluminal assays were performed by employing Transwell system (0.4 um pore size, 0.33 cm$^2$ filter inserts: Transwell®, Costar, Corning). SIINFEKL, VSV-peptide and endotoxin free-, full length ovalbumin (OVA; EndoFit, InvivoGen, San Diego, CA, USA) protein were loaded in the pMBMECs with overnight incubation at 37 °C. Unprocessed peptides were removed by 3x wash with wash buffer containing 10 mM HEPES and 0.1% BSA. Consecutively, $5 \times 10^5$ naïve OT-I cells per well were co-incubated with pMBMECs. Professional Ag presenter, LPS-stimulated BMDCs were co-incubated with the naïve OT-I cells 1:3 ratio (BMDCs/CD8$^+$ T cells) as a positive control for luminal assays. The naïve status of the cells was evaluated before co-incubation by flow cytometry staining with CD25, CD69, CD44 and homing receptor CD62L (Antibodies: Supplementary Table 1). After 24 h, the T cells were collected from pMBMEC- or BMDC co-cultures and were stained for the same markers for the detection of activation. By employing the exact co-culture system, the induction of proliferation of naïve CD8$^+$ T cells was assessed by BrdU Cell proliferation ELISA kit (Abcam, Cambridge, UK) according to the protocol of the manufacturer after 72 h of co-culture. Following the removal of the T cells, pMBMECs were washed and immunofluorescent stained with ZO-1 and DAPI. As an internal control, pMBMECs that were pulsed with peptides but not co-cultured with CD8$^+$ T cells were stained with the same markers to control the T cell unrelated effects.

## In vitro live cell imaging

In vitro live cell imaging of T cell interactions with pMBMECs was performed as described before[45,31,64]. Briefly, prior to the assay WT and B2M$^{-/-}$ pMBMECs are pulsed with either SIINFEKL or VSV peptide (0.5 ng/mL) for 30 min at 37 °C or left unpulsed. Following this step, they were gently washed 3x with wash buffer C (1x HBSS with Ca$^2$ + Mg$^{2+}$ (Gibco), 10 mM HEPES (Gibco), 0.1% BSA) to remove the unprocessed peptide. OT-I or tdTomato OT-I CD8$^+$ T cells ($1 \times 10^6$ cells/ml) were superfused over WT or B2M$^{-/-}$ pMBMECs. Accumulation of CD8$^+$ T cells on pMBMECs in the flow chamber was allowed at a low shear (0.1 dyn/cm$^2$) for 5 min, followed by physiological shear (1.5 dyn/cm$^2$) for an additional 25 min to assess the post-arrest behavior or 55 min to observe the late dynamics. Images were acquired at 10x or 20x magnification with an inverted microscope (AxioObserver, Carl Zeiss, Oberkochen, Germany) with phase-contrast and fluorescence illumination, every 10 s. Image analysis was performed using ImageJ software (ImageJ software, National Institute of Health, Bethesda, USA). Thirty seconds after the onset of the enhanced shear, the number of arrested T cells were counted manually by using ImageJ. The post-arrest behavior of T cells was defined and visualized as fractions of categorized T cells set to 100%, as follows: T cells that actively sent protrusions underneath the endothelium in a stationary position (probing); polarized T cells that continuously crawled on the endothelium (crawling); T cells that transmigrated across the endothelium following probing or crawling (probing + diapedesis, crawling + diapedesis); T cells that did not complete a diapedesis event with prior probing or crawling (probing + partial diapedesis and crawling +

partial diapedesis); T cells that detached during the observation time (detaching). T cells that crawled out of the imaged field of view or went through cell division were counted as arrested but not categorized for the behavior. T cell crawling tracks were evaluated after manual tracking of individual T cells, using the manual tracking plug-in of ImageJ.

## CD8+ T cell-mediated CNS autoimmune disease: ODC-OVA Model

CD8$^+$ T cell mediated autoimmune disease was induced in 9–12-week-old ODC-OVA mice by intravenous injection of freshly isolated naïve tdTomato OT-I cells ($2 \times 10^5$ cells in 100 μl/mouse) and wild-type C57BL/6 J mice were used as controls. 24 h later, mice were peripherally challenged with an intraperitoneal injection of $10^5$ plaque-forming units (PFU) of LCMV-OVA[21]. Animals were monitored twice per day for clinical symptoms and scored as follows: 0, healthy; 0.5, limp tail; 1, hind leg paraparesis; 2, hind leg paraplegia; 3, hind leg paraplegia with incontinence as previously described[21].

## Active EAE

Active EAE (aEAE) was induced in 8–12 weeks old female WT C57BL/6 J using the MOG$_{aa35-55}$-peptide as previously described[63,65]. Pertussis toxin (List Biological Laboratories, Campbell, US) (300 ng in 100 μl PBS/mouse) was applied intraperitoneal (i.p.) on day 0 (immunization day) and day 2 post-immunization. Weights and clinical severity were assessed twice daily and scored as described before[65]: 0, asymptomatic; 0.5, limp tail; 1, hind leg weakness; 2, hind leg paraplegia; 3, hind leg paraplegia and incontinence.

## BBB endothelial cell TAP1-deficient mouse model

Conditional deletion of TAP1 in the brain microvascular endothelium was achieved by intravenous injection of ODC-OVA TAP1$^{floxed/floxed}$ mice with $2 \times 10^{11}$ genomic particles (gp) AAV-BR1-CAG-Cre exactly as previously described[24]. An AAV-BR1-CAG-GFP virus was used as a negative control. Two weeks after AAV-BR1-CAG-Cre or AAV-BR1-CAG-GFP infection, respectively, mice were induced for CD8$^+$ T cell mediated CNS autoimmune disease as described above.

## Intravital microscopy of the cervical spinal cord

Epifluorescence and 2PM intravital microscopy of cervical spinal cord of WT, VE-Cadherin-GFP knock in and ODC-OVA transgenic C57BL/6 J mice was performed exactly as described before[59,66]. This window preparation allows for visualization of subarachnoid, subpial and spinal cord white matter microvessels as described before[67]. Both, naïve and activated OT-I cells were stained with 1 μM Cell-Tracker Green CMFDA dye (Thermo Fisher Scientific) and systemically injected via a carotid artery catheter as described[66]. For 2PM imaging, the blood vessels were in vivo stained with Alexa Fluor 633-conjugated rat-anti mouse endoglin antibody (20 μg/mouse) as described before[66]. Distortion correction of the images during 2PM-imaging was performed by using Vivo Follow 2.0[68]. Sequences of image stacks were transformed into volume-rendered 4D images by Imaris 9.8 software.

Cisterna magna injections of OVA-AF488 during intravital imaging was performed as previously described[69]. In brief, a cannula (SCI: Ref. BB31695-PE/1) mounted to a 29 G needle (Insulin syringe, Terumo or Braun) was inserted into the cisterna magna after the exposure of the dura mater by disection of the neck muscles. The cannula was mounted on a syringe pump (Stoelting, Wood Dale, IL). During 2P-IVM, infusion of 2 μl of OVA-AF488 at a concentration of 1 mg/ml was performed at a speed of 1 μl/min using a syringe pump.

## Statistical analysis

Statistical analysis was performed using GraphPad Prism 7.0 software (La Jolla, CA, USA). Data were compared using various tests as indicated

in each figure legend. Data are presented as mean ± SD or ±SEM and precise p-values are indicated when statistical significance is reached.

**Reporting summary**

Further information on research design is available in the Nature Portfolio Reporting Summary linked to this article.

## Data availability

All data are available in the main text or the supplementary materials. Materials will be made available upon request. Source data are provided with this paper.

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

## Acknowledgements

We thank to Dr. Gaby Enzmann her support with the biosafety level 2 work. Additional thanks go to Charaf Benarafa (IVI, Mittelhäusern, Swit-zerland) for providing the Granzyme B$^{-/-}$ C57BL/6 mice and to Thomas Hünig (Institute of Virology and Immunobiology, Würzburg, Germany) for providing the ODC-OVA C56BL/J mice. This study was funded by grants from Swiss National Science Foundation (grant numbers 31003A_149420 and 310030_189080) to BE, Fondation Pour L'Aide a la Recherche sur la Sclérose en Plaques (ARSEP) to BE and RL, Fondation pour la Recherche Médicale to RL, Swiss National Science Foundation (grant numbers 310030B_201271 and 310030_185321) to DM, European Research Council grant to DM, National Institutes of Health (grant numbers R01 NS103212 and RF1 NS122174) to AAJ.

## Author contributions

Conceptualization: B.E., S.A., J.P. Methodology: S.A., J.P., E.B., N.P., D.M., A.J.J., N.B., R.L., U.D., J.K. and MS. Investigation: S.A., J.P., V.M.S., A.K., T.G., E.B., U.D.. Visualization: S.A., J.P., V.M.S. Funding acquisition: B.E. Supervision: B.E., R.L., M.S.. Writing—original draft: S.A., J.P. Writing—review & editing: S.A., J.P.

## Competing interests

The authors declare no competing interests.
