## [Peer Review File · Nature Communications]

Antigen-recognition detains CD8+ T cells at the blood-brain barrier and contributes to its breakdownThis manuscript has been previously reviewed at another journal that is not operating a transparent peer review scheme. This document only contains reviewer comments and rebuttal letters for versions considered at *Nature Communications*.

REVIEWERS' COMMENTS

Reviewer #1 (expert in cell-cell interactions in the central nervous system)
Absent.

Reviewer #2 (expert in leukocyte trafficking):

This work is elegant, comprehensive and combines both state of the art in vitro and in vivo findings which prove for the first time that brain derived antigen released during autoimmune neuroinflammation is efficiently taken up by perivascular phagocytes and presented on BBB endothelial cells. This presentation slows or prohibits Ag specific CD8 T-cell from entry into the CNS and triggers a CD8 T-cell mediated focal BBB breakdown which is a very important finding.

My only request is textual.

1. In the results section that describes in addition to classical co-stimulatory molecules ICAM-1 and VCAM-1 are elevated on the inflamed CNS vasculature. This elevation can account for the slower crawling of both naïve and effector OT-I shown in Fig. 7. The slower crawling of these CD8 lymphocytes might involve OT-I TCR triggered integrin activation by OVA presented peptides on the CNS vasculature. Another possibility that has to be clarified in the text is that the inflamed CNS of the ODC-OVA mice undergoes remodeling via TCR independent mechanisms, which could involve ICAM-1 and VCAM-1 outside-in signaling. This possibility is hard to prove but should be at least discussed.

Reviewer #3 (expert in multiple sclerosis and neuroinflammation):

The issues raised in my previous critique have been adequately addressed in my opinion.

RESPONSE TO REVIEWERS' COMMENTS

We thank the Reviewers for their feed-back and have outlined our reply to the remaining comment of Reviewer 2 below.

Reviewer #2 (expert in leukocyte trafficking):

This work is elegant, comprehensive and combines both state of the art in vitro and in vivo findings which prove for the first time that brain derived antigen released during autoimmune neuroinflammation is efficiently taken up by perivascular phagocytes and presented on BBB endothelial cells. This presentation slows or prohibits Ag specific CD8 T-cell from entry into the CNS and triggers a CD8 T-cell mediated focal BBB breakdown which is a very important finding.

Answer: We thank the Reviewer for the positive evaluation of our work.

My only request is textual.

1. In the results section that describes in addition to classical co-stimulatory molecules ICAM-1 and VCAM-1 are elevated on the inflamed CNS vasculature. This elevation can account for the slower crawling of both naïve and effector OT-I shown in Fig. 7. The slower crawling of these CD8 lymphocytes might involve OT-I TCR triggered integrin activation by OVA presented peptides on the CNS vasculature. Another possibility that has to be clarified in the text is that the inflamed CNS of the ODC-OVA mice undergoes remodeling via TCR independent mechanisms, which could involve ICAM-1 and VCAM-1 outside-in signaling. This possibility is hard to prove but should be at least discussed.

Answer: We thank the Reviewer for raising this important point which we have addressed by our experimental design and thus can exclude that elevated levels of ICAM-1 and VCAM-1 on the BBB are mediating the reduced crawling speed of the CD8⁺ T cells on the BBB as described here. Please first note that crawling speeds for CD8⁺ T cells in Figure 7 are only reported for in vivo and in vitro activated T cells and are not compared between healthy and diseased animals. In Figure 6 we show that in contrast to VCAM-1 immunostaining for ICAM-1 is not elevated in the CNS vasculature in the ODC-OVA mice during neuroinflammation when compared to the WT controls at the same day after LCMV-OVA infection. Thus, ICAM-1 which has been shown by us and others to mediate T-cell crawling at the BBB is expressed to comparable levels in ODC-OVA and WT control mice. We think that our experimental design verifies that upregulated expression of adhesion molecules on the inflamed BBB is not accountable for the reduction in the crawling speed of the CD8⁺ T cells as observed. Within the ODC-OVA and WT control models we have directly compared the crawling speed of OT-I cells to that of TCRtg CD8⁺ T cells with different antigen specificities, whose cognate antigens are thus not present in ODC-OVA or WT control mice. As we observed reduced crawling speeds solely for OT-I cells specifically in the ODC-OVA context we can conclude that the reduced crawling speed is due to Ag-specific interactions rather than upregulated expression of endothelial adhesion molecules. No decrease of the crawling speed was observed for the control TCR tg CD8⁺ T cells in these experiments in spite of the upregulation

Reviewer #3 (expert in multiple sclerosis and neuroinflammation):

The issues raised in my previous critique have been adequately addressed in my opinion.

Answer: We thank the Reviewer for this positive feed-back.